# WEAKLY SUPERVISED LABEL LEARNING FLOWS

## ABSTRACT

Supervised learning usually requires a large amount of labelled data. However, attaining ground-truth labels is costly for many tasks. Alternatively, weakly supervised methods learn with cheap weak signals that only approximately label some data. Many existing weakly supervised learning methods learn a deterministic function that estimates labels given the input data and weak signals. In this paper, we develop label learning flows (LLF), a general framework for weakly supervised learning problems. Our method is a generative model based on normalizing flows. The main idea of LLF is to optimize the conditional likelihoods of all possible labelings of the data within a constrained space defined by weak signals. We develop a training method for LLF that trains the conditional flow inversely and avoids estimating the labels. Once a model is trained, we can make predictions with a sampling algorithm. We apply LLF to three weakly supervised learning problems. Experiment results show that our method outperforms many state-of-the-art alternatives.

## 1 INTRODUCTION

Machine learning has achieved great success in many supervised learning tasks. However, in practice, data labeling is usually human intensive and costly. To address this problem, practitioners are turning to weakly supervised learning (Zhou, 2018), which trains machine learning models with only noisy labels that are generated by human specified rules or pretrained models for related tasks. Recent research shows that these models trained with weak supervisions can also perform well.

Many existing weakly supervised learning methods (Ratner et al., 2016; 2017; Bach et al., 2019; Arachie & Huang, 2021b) learn a deterministic function that estimates the unknown labels $\mathbf{y}$ given input data $\mathbf{x}$ and weak signals $\mathbf{Q}$. Since the observed information is incomplete, the predictions based on it can be varied. However, these methods ignore this uncertainty between $\mathbf{x}$ and $\mathbf{y}$. In this paper, we develop *label learning flows* (LLF), a general framework for weakly supervised learning problems. LLF is a flow-based generative model (Dinh et al., 2014; Rezende & Mohamed, 2015; Dinh et al., 2016; Kingma & Dhariwal, 2018). The main idea behind LLF is that we define the relationship between $\mathbf{x}$ and $\mathbf{y}$ with a probability distribution $p(\mathbf{y}|\mathbf{x})$, which is modeled by a conditional flow. In training, we use the weak signals $\mathbf{Q}$ to define a constrained space for $\mathbf{y}$ and then optimize the likelihood of all possible $\mathbf{y}$ that are within this constrained space. Therefore, this model captures all possible relationships between the input $\mathbf{x}$ and output $\mathbf{y}$. Learning LLF can be defined as a constrained optimization problem. We develop a learning method for LLF that trains the conditional flow inversely and avoids estimating the labels. For prediction, we use sample-based method (Lu & Huang, 2020) to estimate the labels.

We apply LLF to three weakly supervised learning problems: weakly supervised classification (Arachie & Huang, 2021b; Mazzetto et al., 2021b), weakly supervised regression, and unpaired point cloud completion (Chen et al., 2019; Wu et al., 2020). These three problems have very different label types and weak signals. Our method outperforms all other state-of-the-art methods on weakly supervised classification and regression, and it can perform comparably to other recent methods on unpaired point cloud completion. These experiments show that LLF is versatile and powerful.

## 2 BACKGROUND

In this section, we introduce weakly supervised learning and conditional normalizing flows.

**Weakly Supervised Learning.** Given a dataset $\mathcal{D} = \{\mathbf{x}_1, ..., \mathbf{x}_N\}$, and weak signals $\mathbf{Q}$, weakly supervised learning finds a model that can predict the unknown label $\mathbf{y}_i$ for each input data $\mathbf{x}_i$. Some methods, e.g., ALL (Arachie & Huang, 2021b), define a set of constrained functions based on $\mathbf{Q}$ and $\mathbf{x}$. These functions form a space of possible $\mathbf{y}$. The methods then look for one possible $\mathbf{y}$ within this constrained space. In this work, we follow this idea and use constrained functions to restrict the predicted $\mathbf{y}$.

**Conditional Normalizing Flows.** A normalizing flow (Rezende & Mohamed, 2015) is a series of invertible functions $\mathbf{f} = \mathbf{f}_1 \circ \mathbf{f}_2 \circ \cdots \circ \mathbf{f}_K$ that transform the probability density of output variables $\mathbf{y}$ to the density of latent variables $\mathbf{z}$. In conditional flows (Trippe & Turner, 2018), a flow layer function $\mathbf{f}_i$ is also parameterized by the input variables $\mathbf{x}$, i.e., $\mathbf{f}_i = \mathbf{f}_{x,\phi_i}$, where $\phi_i$ is the parameters of $\mathbf{f}_i$. With the change-of-variable formula, the log conditional distribution $\log p(\mathbf{y}|\mathbf{x})$ can be exactly and tractably computed as

$$\log p(\mathbf{y}|\mathbf{x}) = \log p_Z(\mathbf{f}_{\mathbf{x},\phi}(\mathbf{y})) + \sum_{i=1}^{K} \log \left| \det \left( \frac{\partial \mathbf{f}_{\mathbf{x},\phi_i}}{\partial \mathbf{r}_{i-1}} \right) \right|, \tag{1}$$

where $p_Z(\mathbf{z})$ is a tractable base distribution, e.g. Gaussian distribution. The $\frac{\partial \mathbf{f}_{\mathbf{x},\phi_i}}{\partial \mathbf{r}_{i-1}}$ is the Jacobian matrix of $\mathbf{f}_{\mathbf{x},\phi_i}$. The $\mathbf{r}_i = \mathbf{f}_{\mathbf{x},\phi_i}(\mathbf{r}_{i-1})$, $\mathbf{r}_0 = \mathbf{y}$, and $\mathbf{r}_K = \mathbf{z}$.

To use normalizing flows, we need to develop flow layers that are invertible and have tractable Jacobian determinant. In this paper, we use affine coupling layer (Dinh et al., 2014; 2016) to form normalizing flows. It splits the input to two parts, and force the first part only relate to the second part, so that the Jacobian is a triangular matrix. For conditional flows, we can define conditional affine coupling layer as

$$\begin{aligned}
\mathbf{y}_a, \mathbf{y}_b &= \text{split}(\mathbf{y}), \\
\mathbf{z}_b &= \mathbf{s}(\mathbf{x}, \mathbf{y}_a) \odot \mathbf{y}_b + \mathbf{b}(\mathbf{x}, \mathbf{y}_a), \\
\mathbf{z} &= \text{concat}(\mathbf{y}_a, \mathbf{z}_b),
\end{aligned}$$

where $\mathbf{s}$ and $\mathbf{b}$ are two neural networks. The split() and the concat() functions split and concatenate the variables.

## 3 PROPOSED METHOD

In this section, we introduce the label learning flows (LLF) framework for weakly supervised learning. Given $\mathbf{Q}$ and $\mathbf{x}$, we define a set of constraints to restrict the predicted label $\mathbf{y}$. These constraints can be inequalities, formatting like $f(\mathbf{x}, \mathbf{y}, \mathbf{Q}) \leq b$, or equations, formatting like $f(\mathbf{x}, \mathbf{y}, \mathbf{Q}) = b$. For simplicity, we represent this set of constraints as $\mathbf{C}(\mathbf{x}, \mathbf{y}, \mathbf{Q})$. Let $\Omega$ be the constrained space of all possible $\mathbf{y}$ defined by $\mathbf{C}(\mathbf{x}, \mathbf{y}, \mathbf{Q})$. Previous methods only look for one possible $\mathbf{y}$ within $\Omega$. In contrast, LLF optimizes the conditional log-likelihood of all possible $\mathbf{y}$ within $\Omega$, resulting in the following objective

$$\max_{\phi} \mathbb{E}_{p_{\text{data}}(\mathbf{x})} \mathbb{E}_{\mathbf{y} \sim U(\Omega)} \left[ \log p(\mathbf{y}|\mathbf{x}, \phi) \right], \tag{2}$$

where $U(\Omega)$ is a uniform distribution within $\Omega$, and $p(\mathbf{y}|\mathbf{x})$ is a continuous density model of $\mathbf{y}$.

Let $\hat{\mathbf{y}}$ be the true label. We assume that each data point $\mathbf{x}_i$ only has one unique label $\hat{\mathbf{y}}_i$, so that $p_{\text{data}}(\mathbf{x}, \hat{\mathbf{y}}) = p_{\text{data}}(\mathbf{x})$. Let $q(\hat{\mathbf{y}}|\mathbf{x})$ be a certain model of $\hat{\mathbf{y}}$. Traditional supervised learning learns a $q(\hat{\mathbf{y}}|\mathbf{x})$ that maximizes the cross entropy $\mathbb{E}_{p_{\text{data}}(\mathbf{x}, \hat{\mathbf{y}})} \left[ \log q(\hat{\mathbf{y}}|\mathbf{x}, \phi) \right]$. Following theorem indicates that maximizing $\log p(\mathbf{y}|\mathbf{x})$ can be interpreted as maximizing a lower bound of $\log q(\hat{\mathbf{y}}|\mathbf{x})$.

**Theorem 1** *Let $\Omega^* \subseteq \Omega$ is a tight enough space satisfying that $\hat{\mathbf{y}} \in \Omega^*$ and for two different $\hat{\mathbf{y}}_i$ and $\hat{\mathbf{y}}_j$, the $\Omega_i^*$ and $\Omega_j^*$ are non-overlapped. The volume of $\Omega^*$ is bounded such that $\frac{1}{|\Omega^*|} \leq M$. The relationship between $p(\mathbf{y}|\mathbf{x})$ and $q(\hat{\mathbf{y}}|\mathbf{x})$ can be defined as: $q(\hat{\mathbf{y}}|\mathbf{x}) = \int_{\mathbf{y} \in \Omega^*} p(\mathbf{y}|\mathbf{x})d\mathbf{y}$. Then maximizing $\log p(\mathbf{y}|\mathbf{x})$ can be interpreted as maximizing the lower bound of $\log q(\hat{\mathbf{y}}|\mathbf{x})$. That is,*

$$\mathbb{E}_{p_{data}(\mathbf{x})} \mathbb{E}_{\mathbf{y} \sim U(\Omega^*)} \left[ \log p(\mathbf{y}|\mathbf{x}, \phi) \right] \leq M \mathbb{E}_{p_{data}(\mathbf{x}, \hat{\mathbf{y}})} \left[ \log q(\hat{\mathbf{y}}|\mathbf{x}, \phi) \right] \tag{3}$$

The complete proof is in appendix. Theorem 1 indicates that, when $\Omega$ is well-defined, learning LLF is analogous to dequantization (Theis et al., 2015; Ho et al., 2019). That is, the method optimizes the

likelihood of dequantized true labels. In practice, the real constrained space, i.e., $\Omega$, may be loose and does not fulfill the conditions in Theorem 1, which will result in inevitable errors that come from the weakly supervised setting. Moreover, for some regression problems, the ideal $\Omega^*$ only contains a single point: the ground truth label, i.e., $\Omega^* = \{\hat{\mathbf{y}}\}$.

### 3.1 LEARNING AND PREDICTION

Since $\mathbf{y}$ is unobserved, directly optimizing the conditional likelihood, i.e., Equation 2, is impossible. Using the invertibility of normalizing flows, we can rewrite $\log p(\mathbf{y}|\mathbf{x})$ as

$$
\begin{aligned}
\log p(\mathbf{y}|\mathbf{x}) &= \log p_Z(\mathbf{f}_{\mathbf{x},\phi}(\mathbf{y})) + \sum_{i=1}^{K} \log \left| \det \left( \frac{\partial \mathbf{f}_{\mathbf{x},\phi_i}}{\partial \mathbf{r}_{i-1}} \right) \right| \\
&= \log p_Z(\mathbf{z}) - \sum_{i=1}^{K} \log \left| \det \left( \frac{\partial \mathbf{g}_{\mathbf{x},\phi_i}}{\partial \mathbf{r}_i} \right) \right|,
\end{aligned}
\tag{4}
$$

where $\mathbf{g}_{\mathbf{x},\phi_i} = f_{\mathbf{x},\phi_i}^{-1}$ is the inverse flow.

With the inverse flow, Equation 2 can be interpreted as a constrained optimization problem

$$
\max_{\phi} \mathbb{E}_{p_{data}(\mathbf{x})} \mathbb{E}_{p_Z(\mathbf{z})} \left[ \log p_Z(\mathbf{z}) - \sum_{i=1}^{K} \log \left| \det \left( \frac{\partial \mathbf{g}_{\mathbf{x},\phi_i}}{\partial \mathbf{r}_i} \right) \right| \right],
\tag{5}
$$
$$
\text{s.t.} \quad \mathbf{C}(\mathbf{x}, \mathbf{g}_{\mathbf{x},\phi}(\mathbf{z}), \mathbf{Q}).
$$

Note that in Equation 5, the original constraint for $\mathbf{z}$ is $\mathbf{g}_{\mathbf{x},\phi}(\mathbf{z}) \in \Omega$, and this constraint can be replaced with $\mathbf{C}(\mathbf{x}, \mathbf{g}_{\mathbf{x},\phi}(\mathbf{z}), \mathbf{Q})$. For efficient training, this constrained optimization problem can be approximated with the penalty method, resulting in the objective

$$
\max_{\phi} \mathbb{E}_{p_{data}(\mathbf{x})} \mathbb{E}_{p_Z(\mathbf{z})} \left[ \log p_Z(\mathbf{z}) - \sum_{i=1}^{K} \log \left| \det \left( \frac{\partial \mathbf{g}_{\mathbf{x},\phi_i}}{\partial \mathbf{r}_i} \right) \right| + \lambda \mathbf{C}_r(\mathbf{x}, \mathbf{g}_{\mathbf{x},\phi}(\mathbf{z}), \mathbf{Q}) \right],
\tag{6}
$$

where $\lambda$ is the penalty coefficient, and $\mathbf{C}_r()$ means we reformulate the constraints to be penalty losses. For example, an inequality constraint will be redefined as a hinge loss.

In practice, the expectation with respect to $p_Z(\mathbf{z})$ can be approximated with Monte Carlo estimate with $L$ samples. Since we only need to obtain stochastic gradients, we follow previous works (Kingma & Welling, 2013) and set $L = 1$.

Given a trained model and a data point $\mathbf{x}_i$, prediction requires outputting a label $\mathbf{y}_i$ for $\mathbf{x}_i$. We follow (Lu & Huang, 2020) and use a sample average, i.e., $\mathbf{y}_i = \sum_{j=1}^{L} \mathbf{g}_{\mathbf{x}_i,\phi}(\mathbf{z}_j)$ as the prediction. In our experiments, we found that $L = 10$ is enough for generating high-quality labels.

## 4 CASE STUDY

In this section, we illustrate three scenarios of using LLF to address weakly supervised learning problems.

### 4.1 WEAKLY SUPERVISED CLASSIFICATION

We follow previous works (Arachie & Huang, 2021b; Mazzetto et al., 2021b) and consider binary classification. For each example, the label $\mathbf{y}$ is a two-dimensional vector within a one-simplex. That is, the $\mathbf{y} \in \mathcal{Y} = \{\mathbf{y} \in [0,1]^2 : \sum_i \mathbf{y}[i] = 1\}$, where $\mathbf{y}[i]$ is the $i$th dimension of $\mathbf{y}$. Each ground truth label $\hat{\mathbf{y}} \in \{0,1\}^2$ is a two-dimensional one-hot vector. We have $M$ weak labelers, which will generate $M$ weak signals for each data point $\mathbf{x}_i$, i.e., $\mathbf{q}_i = [\mathbf{q}_{i,1}, ..., \mathbf{q}_{i,M}]$. Each weak signal $\mathbf{q}_{i,m} \in \mathcal{Q} = \{\mathbf{q} \in [0,1]^2 : \sum_i \mathbf{q}[i] = 1\}$ is a soft labeling of the data. In practice, if a weak labeler $m$ fails to label a data point $\mathbf{x}_i$, the $\mathbf{q}_{i,m}$ can be null, i.e., $\mathbf{q}_{i,m} = \emptyset$ (Arachie & Huang, 2021a). Following Arachie & Huang (2021b), we assume we have access to expected error rate bounds of

these weak signals $\mathbf{b} = [\mathbf{b}_1, .., \mathbf{b}_M]$. Therefore, the weak signals imply constraints

$$\sum_{i=1, \mathbf{q}_{i,m} \neq \emptyset}^{N} (1 - \mathbf{y}_i) \odot \mathbf{q_{i,m}} + \mathbf{y}_i \odot (1 - \mathbf{q}_{i,m}) \leq N_m \mathbf{b}_m \quad \forall m \in \{1, ..., M\}, \tag{7}$$

where $N_m$ is the number of data points that are labeled by weak labeler $m$.

This problem can be solved with LLF, i.e., Equation 5, by defining $\mathbf{C}(\mathbf{x}, \mathbf{g}_{\mathbf{x}, \phi}(\mathbf{z}, \mathbf{Q}))$ to be a combination of weak signal constraints, i.e., Equation 7, and simplex constraints, i.e., $\mathbf{y} \in \mathcal{Y}$. The objective function of LLF for weakly supervised classification is

$$\max_{\phi} \quad \log p_Z(\mathbf{z}) - \sum_{i=1}^{K} \log \left| \det \left( \frac{\partial \mathbf{g}_{\mathbf{x}, \phi_i}}{\partial \mathbf{r}_i} \right) \right|$$

$$+ \lambda_1 [\mathbf{g}_{\mathbf{x}, \phi}(\mathbf{z})]_+^2 + \lambda_2 [1 - \mathbf{g}_{\mathbf{x}, \phi}(\mathbf{z})]_+^2 + \lambda_3 \left( \sum_i \mathbf{g}_{\mathbf{x}, \phi}(\mathbf{z})[i] - 1 \right)^2$$

$$+ \lambda_4 \sum_{m=1}^{M} \left[ \sum_{\substack{i=0 \\ \mathbf{q}_{i,m} \neq \emptyset}}^{N} (1 - \mathbf{g}_{\mathbf{x}, \phi}(\mathbf{z})_i) \odot \mathbf{q}_{i,m} + \mathbf{g}_{\mathbf{x}, \phi}(\mathbf{z})_i \odot (1 - \mathbf{q}_{i,m}) - N_m \mathbf{b}_m \right]_+^2, \tag{8}$$

where the second row describes the simplex constraints, and the last row is the weak signal constraints. The $[.]_+$ is a hinge function that returns its input if positive and zero otherwise. We omit the expectation terms for simplicity.

## 4.2 WEAKLY SUPERVISED REGRESSION

For weakly supervised regression, we predict one-dimensional continuous labels $y \in [0, 1]$ given input dataset $\mathcal{D} = \{\mathbf{x}_1, ..., \mathbf{x}_N\}$ and weak signals $\mathbf{Q}$. We define the weak signals as follows. For the $m$th feature of input data, we have access to a threshold $\epsilon_m$, which splits $\mathcal{D}$ to two parts, i.e., $\mathcal{D}_1, \mathcal{D}_2$, such that for each $\mathbf{x}_i \in \mathcal{D}_1$, the $\mathbf{x}_{i,m} \geq \epsilon_m$, and for each $\mathbf{x}_j \in \mathcal{D}_2$, the $\mathbf{x}_{j,m} < \epsilon_m$. We also have access to estimated values of labels for subsets $\mathcal{D}_1$ and $\mathcal{D}_2$, i.e., $b_{m,1}$ and $b_{m,2}$. This design of weak signals tries to mimic that in practical scenarios, human experts can design rule-based methods for predicting labels for given data. For example, in disease prediction, a medical doctor can predict the disease rates for patients based on their ages. For a group of people whose age is greater than a threshold, an experienced physician would know an estimate of their average disease rate. Assuming that we have $M$ rule-based weak signals, the constraints can be mathematically defined as follows:

$$\frac{1}{|\mathcal{D}_{m,1}|} \sum_{i \in \mathcal{D}_{m,1}} y_i = b_{m,1}, \quad \frac{1}{|\mathcal{D}_{m,2}|} \sum_{j \in \mathcal{D}_{m,2}} y_j = b_{m,2}, \quad m \in 1, ..., M. \tag{9}$$

Plugging in Equation 9 to Equation 6, we have

$$\max_{\phi} \quad \log p_Z(z) - \sum_{i=1}^{K} \log \left| \det \left( \frac{\partial \mathbf{g}_{\mathbf{x}, \phi_i}}{\partial r_i} \right) \right| + \lambda_1 [\mathbf{g}_{\mathbf{x}, \phi}(z)]_+^2 + \lambda_2 [1 - \mathbf{g}_{\mathbf{x}, \phi}(z)]_+^2$$

$$+ \lambda_3 \sum_{m=1}^{M} \left( \frac{1}{|\mathcal{D}_{m,1}|} \sum_{i \in \mathcal{D}_{m,1}} \mathbf{g}_{\mathbf{x}, \phi}(z)_i - b_{m,1} \right)^2 + \left( \frac{1}{|\mathcal{D}_{m,2}|} \sum_{j \in \mathcal{D}_{m,2}} \mathbf{g}_{\mathbf{x}, \phi}(z)_j - b_{m,2} \right)^2 \tag{10}$$

## 4.3 UNPAIRED POINT CLOUD COMPLETION

Unpaired point cloud completion (Chen et al., 2019; Wu et al., 2020) is a practical problem in 3D scanning. Given a set of partial point clouds $\mathcal{X}_p = \{\mathbf{x}_1^{(p)}, ..., \mathbf{x}_N^{(p)}\}$, and a set of complete point clouds $\mathcal{X}_c = \{\mathbf{x}_1^{(c)}, ..., \mathbf{x}_N^{(c)}\}$, we want to restore each $\mathbf{x}_i^{(p)} \in \mathcal{X}_p$. Each point cloud is a set of points,

i.e., $\mathbf{x}_i = \{\mathbf{x}_{i,1}, ..., \mathbf{x}_{i,T}\}$, where each $\mathbf{x}_{i,t} \in \mathcal{R}^3$ is a 3D point, and the counts $T$ represent the number of points in a point cloud.

Note that the point clouds in $\mathcal{X}_p$ and $\mathcal{X}_c$ are *unpaired*, so directly modeling the relationship between $\mathbf{x}^{(c)}$ and $\mathbf{x}^{(p)}$ is impossible. This problem can be interpreted as a weakly supervised learning problem, in which the weak signals $\mathbf{Q}$ are derived from the referred complete point clouds $\mathcal{X}_c$. We predict complete point clouds $\mathbf{y} \in \mathcal{Y}$ for partial point clouds in $\mathcal{X}_p$. The conditional distribution $p(\mathbf{y}|\mathbf{x}_p)$ is an exchangable distribution. We follow previous works (Yang et al., 2019; Klokov et al., 2020) and use De Finetti's representation theorem and variational inference to compute its lower bound as the objective.

$$\log p(\mathbf{y}|\mathbf{x}_p) \;\; \geq \;\; \mathbb{E}_{q(\mathbf{u}|\mathbf{x}_p)} \left[ \sum_{i=1}^{T_c} \log p(\mathbf{y}_i|\mathbf{u}, \mathbf{x}_p) \right] - \mathrm{KL}(q(\mathbf{u}|\mathbf{x}_p)||p(\mathbf{u})), \tag{11}$$

where $q(\mathbf{u}|\mathbf{x}_p)$ is a variational distribution of latent variable $\mathbf{u}$. In practice, it can be represented by an encoder, and uses the reparameterization trick (Kingma & Welling, 2013) to sample $\mathbf{u}$. The $p(\mathbf{u})$ is a standard Gaussian prior. The $p(\mathbf{y}_i|\mathbf{u}, \mathbf{x}_p)$ is defined by a conditional flow. We follow Chen et al. (2019); Wu et al. (2020) and use the adversarial loss and Hausdorff distance loss as constraints. The final objective function is

$$\max_{\phi} \quad \mathbb{E}_{q(\mathbf{u}|\mathbf{x}_p)} \left[ \sum_{t=1}^{T_c} \log p_Z(\mathbf{z_t}) - \sum_{i=1}^{K} \log \left| \det \left( \frac{\partial \mathbf{g}_{\mathbf{u},\mathbf{x}_p,\phi_i}}{\partial \mathbf{r}_{t,i}} \right) \right| \right] - \mathrm{KL}(q(\mathbf{u}|\mathbf{x}_p)||p(\mathbf{u}))$$

$$+ \mathbb{E}_{q(\mathbf{u}|\mathbf{x}_p)} \left[ \lambda_1 (D(\mathbf{g}_{\mathbf{u},\mathbf{x}_p,\phi}(\mathbf{z})) - 1)^2 + \lambda_2 d_H(\mathbf{g}_{\mathbf{u},\mathbf{x}_p,\phi}(z), \mathbf{x}_p) \right], \tag{12}$$

where $D()$ represents the discriminator of a least square GAN (Mao et al., 2017), and $d_H()$ represents the Haudorsff distance, which measures the distance between a generated complete point cloud its corresponding input partial point cloud. For clarity, we use $\mathbf{z}_t$ and $\mathbf{r}_t$ to represent variables of the $t$th point in a point cloud, and $\mathbf{g}_{\mathbf{u},\mathbf{x}_p,\phi}(\mathbf{z})$ to represent a generated point cloud. Detailed derivations of Equation 12 are in appendix.

Training with Equation 12 is different from the previous settings, because we also need to train the discriminator of the GAN. The objective for $D()$ is

$$\min_{D} \mathbb{E}_{p_{data}(\mathbf{x}_c)} \left[ (D(\mathbf{x}_c) - 1)^2 \right] + \mathbb{E}_{p_{data}(\mathbf{x}_p), p_Z(\mathbf{z}), q(\mathbf{u}|\mathbf{x}_p)} \left[ D(\mathbf{g}_{\mathbf{x}_p,\mathbf{u},\phi}(\mathbf{z}))^2 \right]. \tag{13}$$

The training process is similar to traditional GAN training. The inverse flow $\mathbf{g}_{\mathbf{u},\mathbf{x}_p,\phi}$ can be roughly seen as the generator. In training, we train the flow to optimize Equation 12 and the discriminator to optimize Equation 13, alternatively.

## 5 RELATED WORK

In this section, we introduce the research that most related to our work.

**Weakly Supervised Learning.** For weakly supervised classification, we use the same strategy as adversarial label learning (ALL) (Arachie & Huang, 2021b) to define constraint functions based on weak signals. ALL then uses a min-max optimization to learn the model parameters and estimate $\mathbf{y}$ alternatively. In contrast to ALL, LLF is a generative model, so it can learn the model parameters and output $\mathbf{y}$ simultaneously, and it does not need a min-max optimization. Moreover, LLF optimizes the likelihoods of all possible $\mathbf{y}$s within $\Omega$, while ALL only estimates one possible $\mathbf{y}$. Other methods also constrain the label space of the predicted labels using weak supervision (Arachie & Huang, 2021a;b; Mazzetto et al., 2021a;b). These methods are deterministic and developed for classification tasks. However, LLF can be used for regression and uses sampling during inference.

Non-constraint based weak supervision methods typically assume a joint distribution for the weak signals and the true labels of the data. These methods use a latent variable model to estimate the labels of the data while accounting for the dependency among the weak signals (Ratner et al., 2016; 2019; Fu et al., 2020). Like these methods, we assume a family of distributions for the label space of the data. This space is defined by the constraints of the weak supervision and the data. Unlike these methods, we use a flow network rather than a graphical model to solve for the label of the

data. Additionally, we do not solve for the dependence amongst the weak signals thereby avoiding the need for making extra assumptions.

**Normalizing Flows.** Normalizing flows (Dinh et al., 2014; Rezende & Mohamed, 2015; Dinh et al., 2016; Kingma & Dhariwal, 2018) have gained recent attention because of their advantages of exact latent variable inference and log-likelihood evaluation. Specifically, conditional normalizing flows have been widely applied to many supervised learning problems (Trippe & Turner, 2018; Lu & Huang, 2020; Lugmayr et al., 2020; Pumarola et al., 2020) and semi-supervised classification (Atanov et al., 2019; Izmailov et al., 2020). However, normalizing flows have not previously been applied to weakly supervised learning problems.

Our inverse training method for LLF is similar to that of injective flows (Kumar et al., 2020). Injective flows are used to model unconditional datasets. They use an encoder network to map the input data $\mathbf{x}$ to latent code $\mathbf{z}$, and then they use an inverse flow to map $\mathbf{z}$ back to $\mathbf{x}$, resulting in an autoencoder architecture. Different from injective flow, LLF directly samples $\mathbf{z}$ from a prior distribution and uses a conditional flow to map $\mathbf{z}$ back to $\mathbf{y}$ conditioned on $\mathbf{x}$. We use constraint functions to restrict $\mathbf{y}$ to be valid, so that does not need an encoder network.

**Point Cloud Modeling.** Recently, Yang et al. (2019) and Tran et al. (2019) combine normalizing flows with variational autoencoders (Kingma & Welling, 2013) and developed continuous and discrete normalizing flows for point clouds. The basic idea of point normalizing flows is to use a conditional flow to model each point in a point cloud. The conditional flow is conditioned on a latent variable generated by an encoder. To guarantee exchangeability, the encoder uses a PointNet (Qi et al., 2017) to extract features from input point clouds.

The unpaired point cloud completion problem is first defined by Chen et al. (2019). They develop pcl2pcl—a GAN (Goodfellow et al., 2014) based model—to solve it. Their method is two-staged. In the first stage, it trains autoencoders to map partial and complete point clouds to their latent space. In the second stage, a GAN is used to transform the latent features of partial point clouds to latent features of complete point clouds. In their follow-up paper (Wu et al., 2020), they develop a variant of pcl2pcl, called multi-modal pcl2pcl (mm-pcl2pcl), which incorporates random noise to the generative process, so that can capture the uncertainty in reasoning.

In contrast to pcl2pcl, LLF can be trained end-to-end. When applying LLF to this problem, LLF has a similar framework to VAE-GAN (Larsen et al., 2016). The main differences are that LLF models a conditional distribution of points, and its encoder is a point normalizing flow.

## 6 EMPIRICAL STUDY

In this section, we evaluate LLF on the three weakly supervised learning problems.

**Model architecture.** For weakly supervised classification and unpaired point cloud completion, the labels $\mathbf{y}$ are multi-dimensional variables. We follow Klokov et al. (2020) and use flows with only conditional coupling layers. We use the same method as Klokov et al. (2020) to define the conditional affine layer. Each flow model contains 8 flow steps. For unpaired point cloud completion, each flow step has 3 coupling layers. For weakly supervised classification, each flow step has 2 coupling layers. For weakly supervised regression, since $y$ is a scalar, we use simple conditional affine transformation as flow layer, which is defined as:$y = \mathbf{s}(\mathbf{x}) * z + \mathbf{b}(\mathbf{x})$, where $\mathbf{s}$ and $\mathbf{b}$ are two neural networks that take $\mathbf{x}$ as input and output parameters for $y$. The flow for this problem contains 8 conditional affine transformations.

For the unpaired point cloud completion task, we need to also use an encoder network, i.e., $q(\mathbf{u}|\mathbf{x}_p)$ and a discriminator $D()$. We follow Klokov et al. (2020); Wu et al. (2020) and use PointNet (Qi et al., 2017) in these two networks to extract features for point clouds.

**Experiment setup.** In weakly supervised classification and regression experiments, we assume that the ground truth labels are inaccessible, so tuning hyper-parameters for models are impossible. We use default settings for all hyper-parameters of LLF, e.g., $\lambda$s and learning rates. We fix $\lambda = 10$ and use Adam (Kingma & Ba, 2014) with default settings, i.e., $\eta = 0.001$, $\beta_1 = 0.9$ and $\beta_2 = 0.999$. We use an exponential learning rate scheduler with a decreasing rate of $0.996$ to guarantee convergence. We track the decrease of loss and when the decrease is small enough, the training stops. Following previous works (Arachie & Huang, 2021b;a), we use full gradient optimization

to train the models. For fair comparison, we run each experiment 5 times with different random seeds $\{0, 10, 100, 123, 1234\}$. We split each dataset to training, simulation, and test sets. We follow Arachie & Huang (2021b;a) and create weak signals with randomly chosen features, and estimate error bounds and thresholds on simulation set.

For experiments with unpaired point cloud completion, the datasets contain validation sets, so we tune the hyper-parameters using these. We use Adam with an initial learning rate $\eta = 0.0001$ and default $\beta$s. The best coefficients for the constraints in Equation 12 are $\lambda_1 = 10, \lambda_2 = 100$. We use stochastic optimization to train the models, and the batch size is 32. Each model is trained for at most 2000 epochs.

## 6.1 WEAKLY SUPERVISED CLASSIFICATION

**Datasets.** We follow Arachie & Huang (2021a;b) and conduct experiments on 12 datasets. Specifically, the Breast Cancer, OBS Network, Cardiotocography, Clave Direction, Credit Card, Statlog Satellite, Phishing Websites, Wine Quality are tabular datasets from UCI repository (Dua & Graff, 2017). The Fashion-MNIST (Xiao et al., 2017) is an image set with 10 classes of clothing types. We choose 3 pairs of classes, i.e., dresses/sneakers (DvK), sandals/ankle boots (SvA), and coats/bags (CvB), to conduct binary classification. We follow Arachie & Huang (2021b) and create 3 synthetic weak signals for each dataset. The IMDB (Maas et al., 2011), SST (Socher et al., 2013) and YELP are real text datasets. We follow Arachie & Huang (2021a) and use keyword-based weak supervision. Each dataset has more than 10 weak signals. For the experiments on tabular datasets, we set the maximum epochs to 2000. Since the experiments on real text datasets are larger, we set the maximum epochs to 500.

**Baselines.** We compare our method with state-of-the-art methods for weakly supervised classification. For the experiments on tabular datasets and image sets, we use ALL (Arachie & Huang, 2021b), generalized expectation (GE) (Druck et al., 2008; Mann & McCallum, 2010) and averaging of weak signals (AVG). For experiments on text datasets, we use CLL (Arachie & Huang, 2021a), Snorkel MeTaL (Ratner et al., 2019), Data Programming (DP) (Ratner et al., 2016), regularized minimax conditional entropy for crowdsourcing (MMCE) (Zhou et al., 2015), and majority-vote. We also show supervised learning (SL) results for reference.

**Results.** We report the mean and standard deviation of accuracy on test sets in Table 1 and Table 2. LLF outperforms all baseline methods on almost all datasets. On some datasets, LLF can perform as well as supervised learning methods. These results prove that LLF is powerful and effective. In our experiments, we also found that the performance of LLF will also be impacted by different initialization of weights. This is why LLF has relatively larger variance on some datasets.

Table 1: Test set accuracy on tabular and image datasets. We report the mean accuracy of 5 experiments, and the subscripts are standard deviation. LLF outperforms other baselines on 10 datasets.

|  | LLF | ALL | GE | AVG | SL |
|---|---|---|---|---|---|
| Fashion MNIST (DvK) | $\mathbf{1.000}_{0.000}$ | $0.995_{0.000}$ | $0.979_{0.000}$ | $0.835_{0.000}$ | $1.000_{0.000}$ |
| Fashion MNIST (SvA) | $\mathbf{0.944}_{0.001}$ | $0.908_{0.000}$ | $0.501_{0.000}$ | $0.791_{0.000}$ | $0.972_{0.000}$ |
| Fashion MNIST (CvB) | $\mathbf{0.916}_{0.038}$ | $0.805_{0.000}$ | $0.501_{0.000}$ | $0.740_{0.000}$ | $0.988_{0.000}$ |
| Breast Cancer | $\mathbf{0.968}_{0.008}$ | $0.937_{0.019}$ | $0.933_{0.016}$ | $0.911_{0.023}$ | $0.973_{0.007}$ |
| OBS Network | $0.684_{0.006}$ | $0.691_{0.011}$ | $0.676_{0.010}$ | $\mathbf{0.709}_{0.024}$ | $0.704_{0.032}$ |
| Cardiotocography | $\mathbf{0.931}_{0.010}$ | $0.795_{0.011}$ | $0.663_{0.061}$ | $0.902_{0.047}$ | $0.941_{0.008}$ |
| Clave Direction | $\mathbf{0.858}_{0.017}$ | $0.750_{0.013}$ | $0.756_{0.028}$ | $0.707_{0.003}$ | $0.963_{0.001}$ |
| Credit Card | $\mathbf{0.680}_{0.022}$ | $0.678_{0.021}$ | $0.492_{0.088}$ | $0.602_{0.010}$ | $0.717_{0.031}$ |
| Statlog Satellite | $\mathbf{0.997}_{0.002}$ | $0.959_{0.008}$ | $0.987_{0.012}$ | $0.915_{0.011}$ | $0.999_{0.001}$ |
| Phishing Websites | $\mathbf{0.906}_{0.003}$ | $0.896_{0.005}$ | $0.870_{0.009}$ | $0.848_{0.002}$ | $0.929_{0.001}$ |
| Wine Quality | $\mathbf{0.647}_{0.017}$ | $0.623_{0.000}$ | $0.445_{0.014}$ | $0.555_{0.000}$ | $0.685_{0.000}$ |

**Ablation Study.** We can directly train the model using only the constraints as the objective function. In our experiments, we found that training LLF without likelihood (LLF-w/o-nll) will still work. However, the model performs worse than training with likelihood. We believe that this is because the likelihood helps accumulate more probability mass to the constrained space $\Omega$, so the model will more likely generate $\mathbf{y}$ samples within $\Omega$, and the predictions are more accurate.

Table 2: Test set accuracy on real text datasets. LLF outperforms all other baselines.

| | LLF | CLL | MMCE | DP | MV | MeTaL | SL |
|---|---|---|---|---|---|---|---|
| SST | $\mathbf{0.766_{0.002}}$ | $0.729_{0.001}$ | $0.727$ | $0.720_{0.001}$ | $0.720_{0.001}$ | $0.728_{0.001}$ | $0.792_{0.001}$ |
| IMDB | $\mathbf{0.804_{0.000}}$ | $0.740_{0.005}$ | $0.551$ | $0.623_{0.007}$ | $0.724_{0.004}$ | $0.742_{0.004}$ | $0.820_{0.003}$ |
| YELP | $\mathbf{0.861_{0.000}}$ | $0.840_{0.001}$ | $0.680$ | $0.760_{0.005}$ | $0.798_{0.007}$ | $0.780_{0.002}$ | $0.879_{0.001}$ |

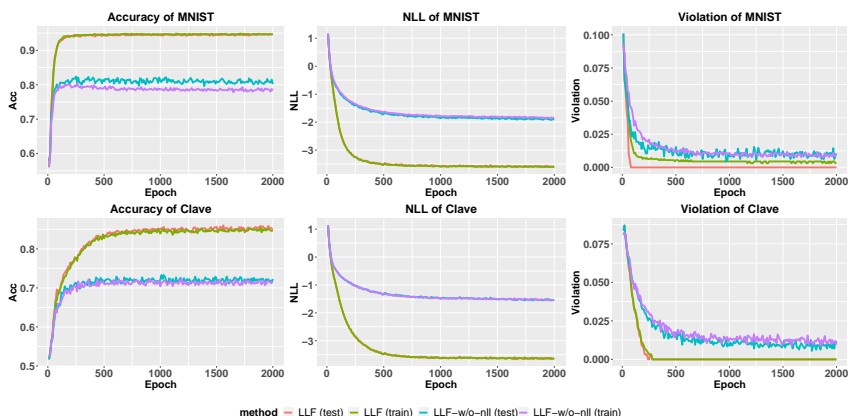

Figure 1: Evolution of accuracy, likelihood and violation of weak signal constraints. Training with likelihood makes LLF accumulate more probability mass to the constrained space, so that the generated $\mathbf{y}$ are more likely to be within $\Omega$, and the predictions are more accurate.

## 6.2 WEAKLY SUPERVISED REGRESSION

**Datasets.** We use 3 tabular datasets from the UCI repository Dua & Graff (2017): Air Quality, Temperature Forecast, and Bike Sharing dataset. For each dataset, we randomly choose 5 features to develop the rule based weak signals. We split each dataset to training, simulation, and test sets. The simulation set is then used to compute the threshold $\epsilon$s, and the estimated label values $b$s. Since we do not have human experts to estimate these values, we use the mean value of a feature as its threshold, i.e., $\epsilon_m = \frac{1}{|\mathcal{D}_{\text{valid}}|} \sum_{i \in \mathcal{D}_{\text{valid}}} \mathbf{x}_i[m]$. We then compute the estimated label values $b_{m,1}$ and $b_{m,2}$ based on labels in the valid set. Note that the labels in the simulation set are only used for generating weak signals, simulating human expertise. In training, we still assume that we do not have access to labels. The original label is within an interval $[l_y, u_y]$. We normalize the original label to within $[0, 1]$ by computing $y = (y - l_y)/(u_y - l_y)$. In prediction, we recover the predicted label to original value by computing $y = y(u_y - l_y) + l_y$.

**Baselines.** To the best of our knowledge, there are no methods specifically designed for weakly supervised regression of this form. We use average of weak signals (AVG) and LFF-w/o-nll as baselines. We also report the supervised learning results for reference.

**Results.** We use root square mean error (RSME) as metric. The results of test set are in Table 3. In general, LLF can predict reasonable labels. Its results are much better than AVG or any of the weak signals alone. Similar to the classification results, training LLF without using likelihood will reduce its performance.

Table 3: Test set RMSE of different methods. The numbers in brackets indicate the label's range. LLF outperforms other baselines on all datasets.

| | LLF | LLF-w/o-nll | AVG | SL |
|---|---|---|---|---|
| Air Quality ($0.1847 \sim 2.231$) | $\mathbf{0.211_{0.009}}$ | $0.266_{0.004}$ | $0.373_{0.005}$ | $0.123_{0.002}$ |
| Temperature Forecast ($17.4 \sim 38.9$) | $\mathbf{2.552_{0.050}}$ | $2.656_{0.055}$ | $2.827_{0.027}$ | $1.465_{0.031}$ |
| Bike Sharing ($1 \sim 999$) | $\mathbf{157.348_{0.541}}$ | $162.697_{1.585}$ | $171.338_{1.300}$ | $141.920_{1.280}$ |

## 6.3 UNPAIRED POINT CLOUD COMPLETION

**Datasets.** We use the Partnet (Mo et al., 2019) dataset in our experiments. We follow Wu et al. (2020) and conduct experiments on the 3 largest classes of PartNet: Table, Chair, and Lamp. We treat each class as a dataset, which is split to training, validation, and test sets based on official splits of PartNet. For each point cloud, we remove points of randomly selected parts to create a partial point cloud. We follow Chen et al. (2019); Wu et al. (2020) and let the partial point clouds have 1024 points, and the complete point clouds have 2048 points. We let the latent variable $\mathbf{u}$ of the VAE to be a 128-dimensional vector.

**Metrics.** We follow Wu et al. (2020) and use minimal matching distance (MMD) (Achlioptas et al., 2018), total mutual difference (TMD), and unidirectional Hausdorff distance (UHD) as metrics. MMD measures the quality of generated. A lower MMD is better. TMD measures the diversity of samples. A higher TMD is better. UHD measures the fidelity of samples. A lower UHD is better.

**Baselines.** We compare our method with pcl2pcl (Chen et al., 2019), mm-pcl2pcl (Wu et al., 2020), and LLF-w/o-nll. We use two variants of mm-pcl2pcl. Another variant is called mm-pcl2pcl-im, which is different from the original model in that it jointly trains the encoder of modeling multi-modality and the GAN.

Table 4: Evaluation results on three classes of PartNet. LLF performs comparable to baselines.

| PartNet | Chair | | | Lamp | | | Table | | |
|---|---|---|---|---|---|---|---|---|---|
| | MMD↓ | TMD↑ | UHD↓ | MMD↓ | TMD↑ | UHD↓ | MMD↓ | TMD↑ | UHD↓ |
| LLF | 1.72 | 0.63 | 5.74 | 2.11 | 0.57 | 4.71 | 1.57 | 0.55 | 5.42 |
| LLF-w/o-nll | 1.79 | 0.47 | 5.49 | 2.21 | 0.41 | **4.61** | 1.57 | 0.43 | 5.13 |
| pcl2pcl | 1.90 | 0.00 | **4.88** | 2.50 | 0.00 | 4.64 | 1.90 | 0.00 | **4.78** |
| mm-pcl2pcl | **1.52** | **2.75** | 6.89 | **1.97** | **3.31** | 5.72 | **1.46** | **3.30** | 5.56 |
| mm-pcl2pcl-im | 1.90 | 1.01 | 6.65 | 2.55 | 0.56 | 5.40 | 1.54 | 0.51 | 5.38 |

**Results.** We list the test set results in Table 4. In general, pcl2pcl has the best fidelity, i.e., lowest UHD. This is because pcl2pcl is a discriminative model, and it will only predict one certain sample for each input. This is also why pcl2pcl has the worse diversity as measured by TMD. Mm-pcl2pcl has the best diversity. However, we found in our experiments that some samples generated by mm-pcl2pcl are invalid, i.e., they are totally different from the input partial point clouds. Therefore, mm-pcl2pcl has the worse fidelity. LLF scores between pcl2pcl and mm-pcl2pcl. It has better UHD than mm-pcl2pcl, and better TMD and MMD than pcl2pcl. The LLF-w/0-nll has a slightly better UHD than LLF. We believe this is because, without using the likelihood, LLF-w/o-nll is trained directly by optimizing the Hausdorff distance. However, the sample diversity and quality, i.e., TMD and MMD, are worse than LLF. As argued by Yang et al. (2019), the current metrics for evaluating point cloud samples all have flaws, so these scores cannot be treated as hard metrics for evaluating model performance. We therefore visualize some samples in appendix, which show that LLF can generate samples that comparable to mm-pcl2pcl.

## 7 CONCLUSION

In this paper, we propose label learning flows, which represent a general framework for weakly supervised learning. LLF uses a conditional flow to define the conditional distribution $p(\mathbf{y}|\mathbf{x})$, so that can model the uncertainty between input $\mathbf{x}$ and all possible $\mathbf{y}$. Learning LLF is a constrained optimization problem that optimizes the likelihood of all possible $\mathbf{y}$ within the constrained space defined by weak signals. We develop a specific training method to train LLF inversely, avoiding the need of estimating $\mathbf{y}$. We apply LLF to three weakly supervised learning problems, and the results show that our method outperforms many state-of-the-art baselines on the weakly supervised classification and regression problems, and performs comparably to other new methods for unpaired point cloud completion. These results indicate that LLF is a powerful and effective tool for weakly supervised learning problems.

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

## A  PROOF OF THEOREM 1

The proof of Theorem 1 is similar to the proof of dequantization (Theis et al., 2015; Ho et al., 2019). The complete proof is as follows.

*Proof.*

$$
\begin{aligned}
\mathbb{E}_{p_{data}(\mathbf{x})}\mathbb{E}_{\mathbf{y}\sim U(\Omega^*)}\left[\log p(\mathbf{y}|\mathbf{x},\phi)\right] &= \sum_{\mathbf{x}} p_{data}(\mathbf{x})\int_{\mathbf{y}\in\Omega^*}\frac{1}{|\Omega^*|}\log p(\mathbf{y}|\mathbf{x})d\mathbf{y} \\
&\leq M\sum_{\mathbf{x}} p_{data}(\mathbf{x})\log\int_{\mathbf{y}\in\Omega^*} p(\mathbf{y}|\mathbf{x})d\mathbf{y} \\
&= M\sum_{\mathbf{x},\hat{\mathbf{y}}} p_{data}(\mathbf{x},\hat{\mathbf{y}})\log q(\hat{\mathbf{y}}|\mathbf{x}) \\
&= M\mathbb{E}_{p_{data}(\mathbf{x},\hat{\mathbf{y}})}\left[\log q(\hat{\mathbf{y}}|\mathbf{x})\right]
\end{aligned}
\tag{14}
$$

In the second row, we use the properties that $\frac{1}{|\Omega^*|}\leq M$, and the Jensen's inequality. In the third row, we use the assumption that $p_{\text{data}}(\mathbf{x})=p_{\text{data}}(\mathbf{x},\hat{\mathbf{y}})$, and the relationship between $p(\mathbf{y}|\mathbf{x})$ and $q(\hat{\mathbf{y}}|\mathbf{x})$.

$\square$

## B  LABEL LEARNING FLOW FOR UNPAIRED POINT CLOUD COMPLETION

In this section, we provide complete derivations of LLF for unpaired point cloud completion. The conditional likelihood $\log p(\mathbf{y}|\mathbf{x}_p)$ is an exchangeable distribution. We use De Finetti's representation theorem and variational inference to derive a tractable lower bound for it.

$$
\begin{aligned}
\log p(\mathbf{y}|\mathbf{x}_p) &= \int p(\mathbf{y},\mathbf{u}|\mathbf{x}_p)d\mathbf{u} \\
&= \int p(\mathbf{y}|\mathbf{u},\mathbf{x}_p)p(\mathbf{u})d\mathbf{u} \\
&\geq \mathbb{E}_{q(\mathbf{u}|\mathbf{x}_p)}\left[\log p(\mathbf{y}|\mathbf{u},\mathbf{x}_p)\right] - \mathrm{KL}(q(\mathbf{u}|\mathbf{x}_p)||p(\mathbf{u})) \\
&\geq \mathbb{E}_{q(\mathbf{u}|\mathbf{x}_p)}\left[\sum_{i=1}^{T_c}\log p(\mathbf{y}_i|\mathbf{u},\mathbf{x}_p)\right] - \mathrm{KL}(q(\mathbf{u}|\mathbf{x}_p)||p(\mathbf{u})), \quad (15)
\end{aligned}
$$

where in the third row, we use Jensen's inequality to compute the lower bound, and in the last row, we use De Finetti's theorem to factorize $p(\mathbf{y}|\mathbf{u},\mathbf{x}_p)$ to the distributions of points.

The least square GAN discriminator and the Hausdorff distance for generated complete point clouds can be treated as two equality constraints

$$
\begin{aligned}
D(\mathbf{y}) &= 1 \\
d_H(\mathbf{y},\mathbf{x}_p) &= 0.
\end{aligned}
$$

Note that the $d_H()$ is non-negative. Convert these two constraints to penalty functions, we have

$$
\begin{aligned}
\max_{\phi} \quad & \mathbb{E}_{q(\mathbf{u}|\mathbf{x}_p)}\left[\sum_{t=1}^{T_c}\log p_Z(\mathbf{z_t}) - \sum_{i=1}^{K}\log\left|\det\left(\frac{\partial \mathbf{g}_{\mathbf{u},\mathbf{x}_p,\phi_i}}{\partial \mathbf{r}_{t,i}}\right)\right|\right] - \mathrm{KL}(q(\mathbf{u}|\mathbf{x}_p)||p(\mathbf{u})) \\
& + \mathbb{E}_{q(\mathbf{u}|\mathbf{x}_p)}\left[\lambda_1(D(\mathbf{g}_{\mathbf{u},\mathbf{x}_p,\phi}(\mathbf{z}))-1)^2 + \lambda_2 d^{HL}(\mathbf{g}_{\mathbf{u},\mathbf{x}_p,\phi}(z),\mathbf{x}_p)\right]. \quad (16)
\end{aligned}
$$

## C  EXPERIMENT DETAILS

In this section, we provide more details on our experiments to help readers reproduce our results.

### C.1  MODEL ARCHITECTURE

For experiments of weakly supervised classification, and unpaired point cloud completion, we use normalizing flows with only conditional affine coupling layers Klokov et al. (2020). Each layer is defined as

$$
\begin{aligned}
\mathbf{y}_a, \mathbf{y}_b &= \mathrm{split}(\mathbf{y}) \\
\mathbf{s} &= \mathbf{m}_s(\mathbf{w}_y(\mathbf{y}_a) \odot \mathbf{w}_x(\mathbf{x}) + \mathbf{w}_b(\mathbf{x})) \\
\mathbf{b} &= \mathbf{m}_b(\mathbf{c}_y(\mathbf{y}_a) \odot \mathbf{c}_x(\mathbf{x}) + \mathbf{c}_b(\mathbf{x})) \\
\mathbf{z}_b &= \mathbf{s} \odot \mathbf{y}_b + \mathbf{b} \\
\mathbf{z} &= \mathrm{concat}(\mathbf{y}_a, \mathbf{z}_b), \quad (17)
\end{aligned}
$$

where $\mathbf{m}, \mathbf{w}, \mathbf{c}$ are all small neural networks.

For LLF, we only need the inverse flow, i.e., $\mathbf{g}_{\mathbf{x},\phi}$, for training and prediction, so in our experiments, we actually define $\mathbf{g}_{\mathbf{x},\phi}$ as the forward transformation, and let $\mathbf{y} = \mathbf{s} \odot \mathbf{z} + \mathbf{b}$. We do this because multiplication and addition are more stable than division and subtraction.

#### C.1.1  WEAKLY SUPERVISED CLASSIFICATION

In this problem, we use a flow with 8 flow steps, and each step has 2 conditional affine coupling layers. These two layers will transform different dimensions. Each $\mathbf{w}$ and $\mathbf{c}$ are small MLPs with two linear layers. Each $\mathbf{m}$ has one linear layer. The hidden dimension of linear layers is fixed to 64.

#### C.1.2  WEAKLY SUPERVISED REGRESSION

In this problem, since the label $y$ is a scalar, we use conditional affine transformation introduced in Section 6, as a flow layer. A flow has 8 flow layers. The $\mathbf{s}$ and $\mathbf{b}$ in a flow layer are three layer MLPs. The hidden dimension of linear layers is 64.

### C.1.3 UNPAIRED POINT CLOUD COMPLETION

The model architecture used LLF used for this problem is illustrated in Figure 2. We use the same architecture as DPF (Klokov et al., 2020) for point flow. Specifically, the flow has 8 flow steps, and each step has 3 conditional affine coupling layers, i.e., Equation 17. Slightly different from the original DPF, the conditioning networks $\mathbf{w}_x$, $\mathbf{c}_x$, $\mathbf{w}_b$, and $\mathbf{c}_b$ will take the latent variable $\mathbf{u}$ and the features of partial point cloud $\mathbf{x}_p$ as input. The $\mathbf{w}$s and $\mathbf{c}$s are MLPs with two linear layers, whose hidden dimension is 64. The $\mathbf{m}$s are one layer MLPs.

We use a PointNet (Qi et al., 2017) to extract features from partial point cloud $\mathbf{x}_p$. Following Klokov et al. (2020), the hidden dimensions of this PointNet is set as $64-128-256-512$. Given the features of $\mathbf{x}_p$, the encoder $E$ then uses the reparameterization trick (Kingma & Welling, 2013) to generate latent variable $\mathbf{u}$. The encoder has two linear layers, whose hidden dimension is 512.

The GAN discriminator uses another PointNet to extract features from (generated) complete point clouds. We follow Wu et al. (2020) and set the hidden dimensions of this PointNet as $64-128-128-256-128$. The discriminator $D$ is a three layer MLP, whose hidden dimensions are $128-256-512$.

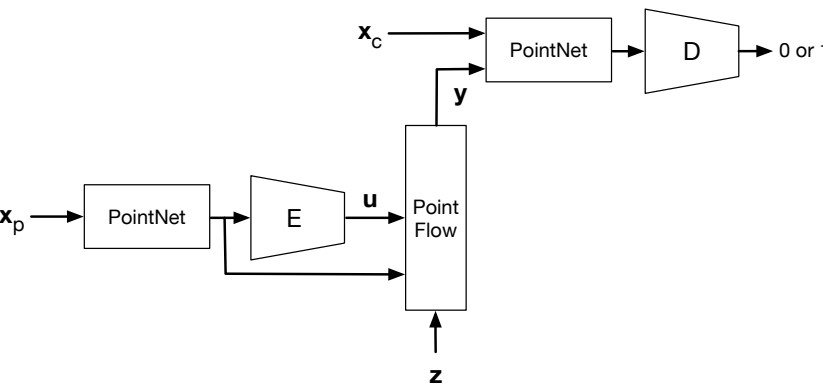

Figure 2: Model architecture of LLF for unpaired point cloud completion. The $E$ represents the encoder, and the $D$ represents the GAN discriminator.

## C.2 MORE EXPERIMENT DETAILS

### C.2.1 WEAKLY SUPERVISED CLASSIFICATION

We use the same way as Arachie & Huang (2021b;a) to split each dataset to training, simulation, and test sets. We use the data and labels in simulation sets to create weak signals, and estimated bounds. We train models on training sets and test model on test sets. We assume that the models do not have access to any labels. The labels in simulation sets are only used to generated weak signals and estimate bounds.

For experiments on Fashion-MNIST and the tabular datasets, we follow Arachie & Huang (2021b) and choose 3 features to create weak signals. We train a logistic regression with each feature on the simulation set, and use the label probabilities predicted by this logistic regression as weak signals. We compute the error of this trained logistic regression on simulation set as estimated error bound.

For experiments on real text datasets, we use the same keyword-based method as Arachie & Huang (2021a) to create weak supervision. Specifically, we choose key words that can weakly indicate positive and negative sentiments. Documents containing possitive words will be labeled as positive, and vice versa.

We list some main features of these datasets in Table 5. We refer to their original papers for more details. For those datasets without official splits, we randomly split them with a ratio of $4:3:3$.

Table 5: Summary of datasets used in weakly supervised classification experiments. The "—" indicates this dataset does not have a official split

| Dataset | Size | Train Size | Test Size | No. features | No. weak signals |
|---------|------|-----------|-----------|--------------|------------------|
| Fashion MNIST (DvK) | $14,000$ | $12,000$ | $2,000$ | 784 | 3 |
| Fashion MNIST (SvA) | $14,000$ | $12,000$ | $2,000$ | 784 | 3 |
| Fashion MNIST (CvB) | $14,000$ | $12,000$ | $2,000$ | 784 | 3 |
| Breast Cancer | 569 | — | — | 30 | 3 |
| OBS Network | 795 | — | — | 21 | 3 |
| Cardiotocography | 963 | — | — | 21 | 3 |
| Clave Direction | $8,606$ | — | — | 16 | 3 |
| Credit Card | $1,000$ | — | — | 24 | 3 |
| Statlog Satellite | $3,041$ | — | — | 36 | 3 |
| Phishing Websites | $11,055$ | — | — | 30 | 3 |
| Wine Quality | $4,974$ | — | — | 11 | 3 |
| IMDB | $49,574$ | $29,182$ | $20,392$ | 300 | 10 |
| SST | $5,819$ | $3,998$ | $1,821$ | 300 | 14 |
| YELP | $55,370$ | $45,370$ | $10,000$ | 300 | 14 |

## C.2.2 WEAKLY SUPERVISED REGRESSION

We use three datasets from the UCI repository. For each dataset, we randomly split it to training, simulation, and test sets with a ratio of $4:3:3$. We use the simulation set to create weak signals and estimated label values. We choose 5 features to create weak signals for each dataset. The detailed introduction of these datasets are as follows. Table 6 summarize the statistical results of them.

**Air Quality.** In this task, we predict the absolute humidity in air, based on other air quality features such as hourly averaged temperature, hourly averaged $NO_2$ concentration etc. The raw dataset has $9,358$ instances. We remove those instances with Nan values, resulting in a dataset with $8,991$ instances. We use hourly averaged concentration CO, hourly averaged Benzene concentration, hourly averaged $NO_x$ concentration, tungsten oxide hourly averaged sensor response, and relative humidity as features for creating weak signals.

**Temperature Forecast.** In this task, we predict the next day maximum air temperature based on current day information. The raw dataset has $7,750$ instances, and we remove those instances with Nan values, resulting in $7,588$ instances. We use present max temperature, forecasting next day wind speed, forecasting next day cloud cover, forecasting next day precipitation, solar radiation as features for creating weak signals.

**Bike Sharing.** In this task, we predict the count of total rental bikes given weather and date information. The raw dataset has $17,389$ instances, and we remove those instances with Nan values, resulting in $17,379$ instances. We use season, hour, if is working day, normalized feeling temperature, and wind speed as features for creating weak signals.

Table 6: Summary of datasets used in weakly supervised regression experiments

| Dataset | Size | No. features | No. weak signals |
|---------|------|--------------|------------------|
| Air Quality | $8,991$ | 12 | 5 |
| Temperature Forecast | $7,588$ | 24 | 5 |
| Bike Sharing | $17,379$ | 12 | 5 |

## C.2.3 UNPAIRED POINT CLOUD COMPLETION

**Datasets.** We use the same way as Wu et al. (2020) to process PartNet Mo et al. (2019). PartNet provides point-wise semantic labels for point clouds. The original point clouds are used as complete point clouds. To generate partial point clouds, we randomly removed parts from complete point

clouds, based on the semantic labels. We use Chair, Table, and Lamp categories. The summary of these three subsets are in Table 7.

Table 7: Summary of datasets used unpaired point cloud completion experiments

| Dataset | Train Size | Valid Size | No. Test Size |
|---------|-----------|-----------|---------------|
| Chair   | $4,489$   | $617$     | $1,217$       |
| Table   | $5,707$   | $843$     | $1,668$       |
| Lamp    | $1,545$   | $234$     | $416$         |

**Metrics.** Let $\mathcal{X}_c$ be the set of referred complete point clouds, and $\mathcal{X}_p$ be the set of input partial point clouds. For each partial point cloud $\mathbf{x}_i^{(p)}$, we generate $M$ complete point cloud samples $\mathbf{y}_i^{(1)}, ..., \mathbf{y}_i^{(M)}$. All these samples form a new set of complete point clouds $\mathcal{Y}$. In our experiments, we follow Wu et al. (2020) and set $M = 10$.

The MMD Achlioptas et al. (2018) is defined as

$$\text{MMD} = \frac{1}{|\mathcal{X}_c|} \sum_{\mathbf{x}_i \in \mathcal{X}_c} d_C(\mathbf{x}_i, \text{NN}(\mathbf{x}_i)), \tag{18}$$

where $\text{NN}(\mathbf{x})$ is the nearest neighbor of $\mathbf{x}$ in $\mathcal{Y}$. The $d_C$ represents Chamfer distance. MMD computes the distance between the set of generated samples and the set of target complete shapes, so it measures the quality of generated.

The TMD is defined as

$$\text{TMD} = \frac{1}{|\mathcal{X}_p|} \sum_{i=1}^{|\mathcal{X}_p|} \left( \frac{2}{M-1} \sum_{j=1}^{M} \sum_{k=j+1}^{M} d_C(\mathbf{y}_i^{(j)}, \mathbf{y}_i^{(k)}) \right). \tag{19}$$

TMD measures the difference of generated samples given an input partial point cloud, so it measures the diversity of samples.

The UHD is defined as

$$\text{UHD} = \frac{1}{|\mathcal{X}_p|} \sum_{i=1}^{|\mathcal{X}_p|} \left( \frac{1}{M} \sum_{j=1}^{M} d_H(\mathbf{x}_i, \mathbf{y}_i^{(j)}) \right), \tag{20}$$

where $d_H$ represents the unidirectional Hausdorff distance. UHD measures the similarity between generated samples and input partial point clouds, so it measures the fidelity of samples.

**Samples.** We compare LLF to mm-pc2pc in Figure 3, and more samples of LLF in Figure 4, Figure 5, and Figure 6. LLF can generate samples that are as good as mm-pc2pc. Mm-pc2pc has a higher diversity in samples, but some generated samples may be unreasonable.

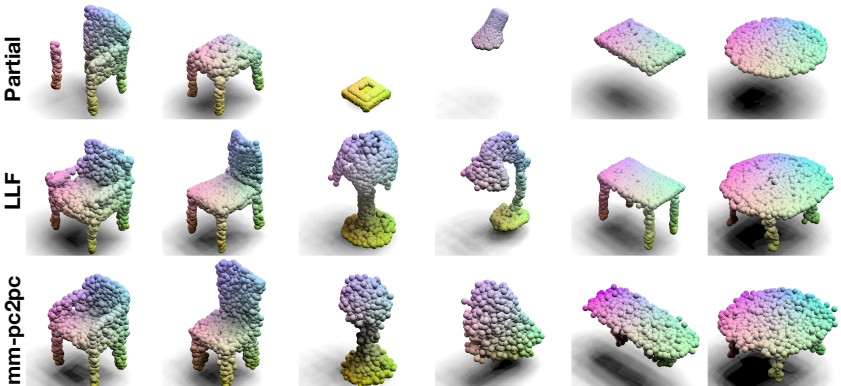

Figure 3: Random sample point clouds generated LLF and mm-pc2pc. The point clouds generated by LLF are as realistic as mm-pc2pc. Mm-pc2pc has a higher diversity in samples. However, sometimes it may generate unreasonable or invalid shapes.

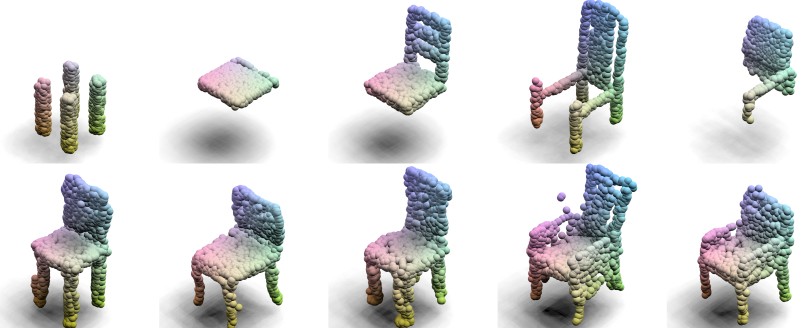

Figure 4: Random chair samples generated by LLF. The first row is partial point clouds, and the second row is generated complete point clouds.

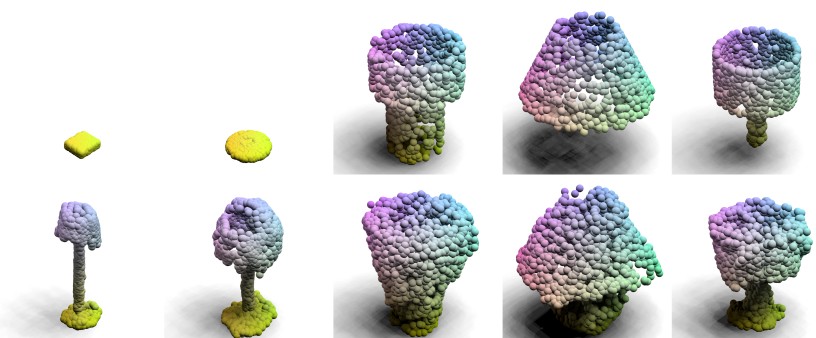

Figure 5: Random lamp samples generated by LLF.

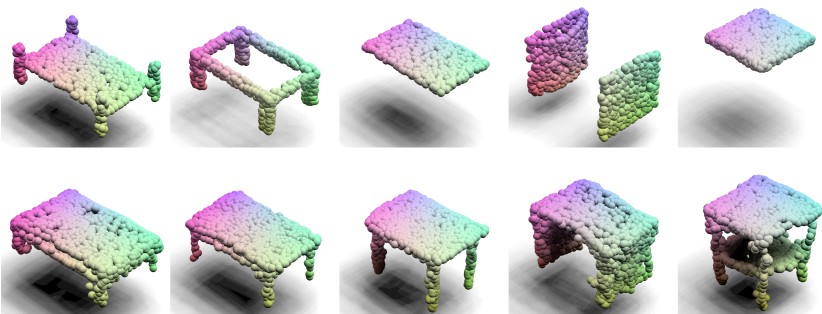

Figure 6: Random table samples generated by LLF.

