# OpenReview forum: "Weakly Supervised Label Learning Flows"
_ICLR.cc/2022/Conference — ICLR 2022 Submitted_

### Official Review · Reviewer_tWTj · 2021-10-21

**Correctness:** 3
**Technical Novelty And Significance:** 3
**Empirical Novelty And Significance:** 4
**Recommendation:** 6
**Confidence:** 4

**Main Review:**

This paper has some positive aspects as follows:
1. The proposed method is novel. This method proposes to optimize the conditional likelihoods of all possible labeling of the data within a constrained space defined by weak signals, which can alleviate the issue that the existing methods ignore the uncertainty between x and y.
2.  The experiment is abundant. The proposed method is validated in three different tasks, including weakly supervised classification, weakly supervised regression and unpaired point cloud completion, which shows its effectiveness.
3.  The paper is well written, the arguments are clear, and the methodology is well presented.

 The authors need to deal with some issues as follows:

4.  In Table 1,  for OBS Network, the performance of LLF is poor than other methods, while on other datasets can obtain the obvious improvement.  the authors should provide an explanation for this result.
5. For the results in Table 4, it is better, if the authors can give a comprehensive evaluation protocol, which can mix MMD, TMD and UHD together, thus it can obviously highlight the superiority of LLF in unpaired point cloud completion.


**Summary Of The Paper:**

Different from many existing weakly supervised learning methods, which learn a deterministic function that estimates labels given the input data and weak signals,  this paper proposes label learning flows (LLF), a general framework for weakly supervised learning problems, and it is a generative model based on normalizing flows. The main idea of LLF is to optimize the conditional likelihoods of all possible labeling of the data within a constrained space defined by weak signals.  The proposed method is applied to three weakly supervised learning problems. Experimental results show that this method outperforms many state-of-the-art alternatives.

**Summary Of The Review:**

The proposed method is novel and the experiment is sufficient. More experimental analysis can further improve the quality of this paper.

---

> ### Author Response · Authors · 2021-11-15
> **Response to Reviewer tWTj**
>
> Thank you for your positive comments!
>
> 1. About experiments on the OBS dataset. There is a bad weak signal in the OBS dataset, which also impacts the performance of other baselines except for the AVG method. Therefore, other baselines are all worse than AVG. The performance of LLF is actually better than GE, and only slightly worse than ALL, so the performance of LLF is still reasonably well. Another important reason is the default setting of hyper-parameters does not fit this dataset well. That is, we tried to tune the hyper-parameters for LLF on this dataset, and got an accuracy of around 0.7, which is slightly better than ALL. However, tuning hyper-parameters violates our assumption for this problem that the labels are not accessible, so we decided to not report this result. We will add more analysis to our camera-ready paper.
>
> 2. About Table 4. Thank you for your suggestions. Actually, unpaired point cloud completion is a new computer vision task, and has not been widely studied. Previous works [1,2] use MMD, TMD, UHD as metrics to measure the quality of generated complete point clouds. We follow them and use the same metrics. However, in our experiments, we found that these metrics have flaws, so we visualized our samples in appendix C.2.3. We also discuss it in the last paragraph of Section 6.3. We will clarify it in our camera-ready paper.
>
>     We agree that developing a comprehensive evaluation protocol is important for this problem. We will discuss it in our main paper, and leave it as future works.
>
>
> [1]  Xuelin Chen et al. Unpaired point cloud completion on real scans using adversarial training.
>
> [2] Rundi Wu et al. Multimodal shape completion via conditional generative adversarial networks.

---

### Official Review · Reviewer_W3Vi · 2021-11-01

**Correctness:** 2
**Technical Novelty And Significance:** 2
**Empirical Novelty And Significance:** 2
**Recommendation:** 5
**Confidence:** 3

**Main Review:**

Major Points:
- Clarity regarding formal definition of the model.  Overall, the paper does not sufficiently describe some important technical details, especially with regards to prior work.  In many places, informal language is used instead of specific, formal definitions of key aspects.  This makes it difficult to fully grasp the work without having read numerous papers on the subject if certain points can be gleaned at all.  I am not suggesting that the work has to extensively review the methods LLF is based on, but the paper could be greatly improved simply by fully elaborating on the construction of the method and taking the time to formally define important concepts.  Some examples include:
1. 3. “In this paper, we use affine coupling layer (Dinh et al., 2014; 2016) to form normalizing flows. It splits the input to two parts, and force the first part only relate to the second part, so that the Jacobian is a triangular matrix”.   The equation below this elaborates further by introducing a “split” function, but it isn’t clear 1) How the vector “y” is split 2) Whether the particular split is important (can you split them any way?) 3) How this results in a triangular Jacobiam 4) Why a triangular Jacobian is important here.
2. “Previous methods only look for one possible y within Ω.” This needs a citation.  What methods?
3. Theorem 1:
    A.  “tight enough space” is too informal.  What is meant here?
    B. It is stated that for two different true labels their constrained spaces are “non-overlapped”:
        I. Does this mean “disjoint”?
        II. It is stated without direct relationship to the rest of the theorem.  Is this assumption?  Is it for all pairs of labels or just two?
        III.  I think the main takeaway from this theorem is to directly compare LLF to a technique that fix labels derived from weak supervision, namely the LLE objective lower bounds the traditional cross-entropy objective, which would be used in the case where weak supervision is used to obtain hard labels for training.  Not only does  I think not only does this point need to be more explicit, but if there is a technique in the literature that does the latter, it should be cited.  Further, the intuitive benefit or insight this theorem shows should be stated more explicitly and clearly.  Later it is written: “learning LLF is analogous to dequantization (Theis et al., 2015; Ho et al., 2019).”.  This is helpful, but elaborating on what dequantization is and why this relationship is important would be better.
4. It’s not clear to me how Equations 2 and 4 result in the constrained optimization in Equation 5.  Is it that you’re simply replacing the sampling assumption in Equation 2 with a constraint?
5. “L” is introduced without definition.  Later, through context, I was able to understand it as the number of samples of z ~ p_Z, but it should be made clearer.
6. Equation 7 and Equation 8: What does the inequality mean between the two vector operands?  Is it element wise?
7. The problem definition of the unpaired point cloud completion problem (Section 4.3) is rather informal.  What specifically does it mean to “restore” an incomplete point cloud?  How is it measured?
8. It is unclear from Section 6 what the various hyper parameters for LLF are set to.  Namely, the paper states that that “\lambda = 10”.  Does this indicate that for all experiments and all different \lambda hyper parameters across all learning problems (this encapsulates 9 different hyper parameters) are set to 10?  If this is true, this is rather remarkable due to the variety of influences each constraint term has on the proposed objectives (discussed later)

- Practicality. Some of the motivation for key assumptions made in the paper lack argumentation for their practicality.  In my mind this hurts the potential impact of the paper.
1. Why is it a practical assumption that the error rate for a weak labeler to be known a priori in the weak classification case?  How does one obtain this in practice?
2. There should be some discussion on the kinds of problems the weakly supervised setting proposed in this work applies to.  The case studies help, but those seem highly specialized and weakly motivated.  For instance, the weakly supervised regression seems somewhat impractical. In general, humans struggle to provide real valued labels such as those suggested int he given example.    I think such single-variate, rule-based assessments make sense for classification problems, where humans can more easily make class assessments based on thresholds (such as “patient has a disease if lab test x_i is over threshold e).   Even in the given example, the underlying problem is better modeled as classification problem (disease versus not-disease), where the labels are noisy assessments of confidence in class membership.

- Other technical questions that need to be addressed:
1. “Since y is unobserved, directly optimizing the conditional likelihood, i.e., Equation 2, is impossible.”  I don’t understand this.  If p(x) and \Omega are given, you should be able to compute Equation 2.  Sampling from p(x) for the outer expectation gives you, x.  Sampling from U(\Omega) for the inner expectation gives you y.  The optimization variable is \phi.  Why can’t I compute p(y|x,\phi)?  Is it rather that sampling within the constraint set is impossible or that it’s simply computationally infeasible in practice?  Clarification is needed.
2. In Equation 8, it seems very important to ensure g(x) is producing valid soft-labels in the simplex and, similarly, in Equation 9 it seems important to ensure the regression labels are in [0.1] (which is an assumption I would like to see explicitly stated in the paper).  The reason I believe this is that other constraint terms could easily push the objective to infinity (namely, the lambda_1 terms).  How is this managed practically?  Would projected gradient ascent be more appropriate to ensure model outputs are on the proper simplex to alleviate this issue?
3. It is difficult to interpret the performance of the proposed methods as well as the baselines if there is assumed to be no ground truth available for validation and all hyperparameters are default.  How can one be sure that the performance of LLF relative to the baselines cannot be explained simply by assuming bad default parameters for the baselines?

Minor Points:
1. “attaining” in the second sentence in the abstract seems like the wrong word.  Perhaps “obtaining”?
2. “Therefore, this model captures all possible relationships between the input x and output y”.  Seems dubious.  Certainly, this is subject to the class of functions used to learn p(y|x).
3. Consider using a different function name for “f” (constraint functions) in section 3, as “\textbf{f}” is used as a the normalizing flow vector function above.  Having them be the same character might make readers try to understand them as directly related, when they are not.
4. I believe in equation 2 “p(y|x,\phi)” is the same as equation 1’s p(y|x).  If so, they should be consistent (either they both should be conditioned on \phi, or neither)


**Summary Of The Paper:**

This work focuses on the problem of learning predictive models where no ground truth labels are provided.  Instead a set of weak constraints over instances that restrict the space of possible target values are provided.  The proposed method, LLF, is based on the core idea that instead of determining a single true weak label for each instance, a model can be learned by considering the expectation over all possible labels in the feasible set defined by the given constraints.  The authors utilize a conditional normalizing flow model as a generative process for labels and constrain it's output by imposing regularization terms on the training objective that represent different constraints imposed by the weak supervision.  The authors show how their framing of weakly supervised learning can be applied go specific settings in classification, regression, and unpaired point cloud completion.  In their evaluation, the authors compare LLF to a number of weakly supervised baselines and show superior performance on classification and regression tasks, while largely being outperformed by one method in point cloud completion.

**Summary Of The Review:**

Overall, the idea of modeling the range of possible labels for instances based on weak supervision instead of simply using a single estimate is a very interesting one.  However, the paper suffers from a number of major issues.  1) The experiments use default hyperparameters for all techniques, thus it is difficult to tell whether the relative performance of methods is indicative of their quality or the quality of the particular default values of their hyperparameters. 2) The normalizing flow component is not well motivated nor is it clearly presented in the context of learning from weak supervision. 3) Some of the proposed used cases lack strong practical motivation.  Many of the major issues I present in the main review seem difficult to remedy in a short period of time, though, if they are the result of a misunderstanding on my end and small edits to the paper could alleviate future readers from similar confusion, I would be willing to change my score.

---

> ### Author Response · Authors · 2021-11-15
> **Response to Reviewer W3Vi**
>
> Thank you for your insightful comments!
>
> Major points
>
> 1. About the affine coupling layers. Affine coupling layers are commonly used flow layers in normalizing flows, and many details of this layer have been well explained by many other papers, e.g., [1,2,3]. Briefly, $\textbf{y}$ is split arbitrarily so that only one half depends on the other. This unidirectional dependency creates a block structure that makes the Jacobian triangular. The triangular Jacobian makes the calculation of the determinant efficient in the change-of-variables formula.
>
> 2. About the previous works. We were referring to ALL [4] and its variants. We will add the appropriate citations.
>
> 3. About the theorem.
>
>    A. We define the “tight enough space” in the next sentence. For two different $\textbf{y}_i$ and $\textbf{y}_j$, their constrained spaces $\Omega_i$ and $\Omega_j$ do not overlap. We will clarify this statement in our camera-ready paper.
>
>    B
>
>    I. “Non-overlapped” means disjoint, and it is an assumption.
>
>    II. It is for any pair of different $\mathbf{y}_i$ and $\mathbf{y}_j$. This assumption can guarantee that the $q(\hat{\mathbf{y}}|\mathbf{x}) = \int p(\mathbf{y}|\mathbf{x}) d\mathbf{y}$ holds, which will be used in the proof in appendix A.
>
>    III. Yes, your understanding is correct. The theorem wants to show a relationship between the LLF’s objective function and the cross-entropy loss. Previous works [5,6] develop a relationship between dequantization and cross-entropy loss. We use a similar method to prove our theorem. We will clarify this theorem in our camera-ready paper.
>
> 4. About the objective function. To derive Equation 5, we replace $log p(\textbf{y}|\textbf{x})$ to the inverse flow in Equation 4, and replace the expectation $E_{\textbf{y} \sim U(\Omega)}[]$ with the constraints $\mathbf{C}$ and another expectation $E_{\mathbf{z} \sim p_Z(\mathbf{z})}[]$.
>
> 5. About the $L$. Thank you for your suggestion, and we will clarify $L$ in our camera-ready paper.
>
> 6. About Equation 7 and Equation 8. The constraint in Equation 7 is directly applied from ALL[1]. The left-hand side of Equation 7 is the total error made by weak signal $\mathbf{q}_{i,m}$ which is bounded by $N_mb_m$.  We are missing an extra index for the classes, i.e., $\mathbf{y}\[k\]_i$, $\mathbf{q}\[k\]_i$, where $k \in \[0,1\]$. With this index, both the left-hand side and the right-hand side are all scalar. We will fix these two equations and make them clearer in the camera-ready version.
>
> 7. About the point cloud completion. Given a partial point cloud, the “restore” means we want to produce its corresponding clean and complete point cloud. We actually use exactly the same setting as [7,8] for this problem. We will explain this problem clearer in our camera-ready paper.
>
> 8. About the $\lambda$. For the weakly supervised classification and regression problems, we set $\lambda=10$. This is because in these two problems, we assume that we don’t have validation sets for tuning hyper-parameters. This is a common assumption in these two problems, and many other papers, e.g., [4], use the same setting. We also want to mention that this default setting of hyper-parameters may not get optimal performance.

---

> > ### Author Response · Authors · 2021-11-15
> > **Response to Reviewer W3Vi  Continue**
> >
> > Practicality.
> >
> > 1. About the assumption of weakly supervised classification. This assumption is proposed in ALL [1]. The argument is that expert annotators who generate labeling functions often have a rough idea how often their rules are valid. Moveover, in the ALL paper, the authors show that their method works with wrong estimates of the bounds. In practice we can estimate the bounds using different methods such as agreement or disagreement rates of the weak signals.
> >
> > 2. About the weakly supervised regression. A discussion of practicality is indeed important. We disagree with the claim that regression does not fit weakly supervised settings. The types of weak signals we use in regression are no more difficult to generate than ones for regression. Human experts will often know a loose average value of a variable within a certain population. We don’t agree that humans necessarily struggle to do this. Consider guessing the average age of a college student, or the average price of a home in San Francisco, or the average monthly electrical power usage for a single-family home. Finally, the example you make about the threshold is exactly the inverse of a regression rule. If a patient is known to have a disease (or a family history of the disease, etc.), we would expect the lab test score to be above threshold e. This type of rule fits exactly into the constraint-based weak supervision framework.
> >
> >
> > Other technical questions
> >
> > 1. About Equation 2. Yes, your understanding is correct. Sampling within the constraint space is impossible, because there is no way to explicitly define the constraint space. We will clarify it in our camera-ready paper.
> >
> > 2. About the objective functions. Thank you for pointing out this problem. These are actually minor mistakes in our equations. In these objective functions, we actually want to maximize the log likelihoods, and minimize the constraint violations at the same time. Therefore, the operators of constraints should have been minus rather than plus. Therefore, in our training, we actually minimize these constraints, and since the values of these constraints are non-negative, by minimizing them in training, we will not get to infinite values. For example, in Equation 8, the term of $\lambda_1$ should be $- \lambda_1 \[ \mathbf{g}( \mathbf{z}) \]_{+}^2$.
> >
> >
> > 3. About the experiments of weakly supervised classification and regression. For weakly supervised classification, we use the same methods as other papers to process the datasets, generate weak signals and set the hyper-parameters for baselines, in order to guarantee that the baseline methods perform the same as stated in their papers. Actually, the accuracies of baselines we report in our paper are almost the same as the accuracies reported in the original papers, which can prove that these baselines achieve their best performance. For weakly supervised regression, the AVG calculates the mean of weak signals as prediction, so it does not have parameters.
> >
> > Minor points.
> >
> > 1. About the typos and bugs. Thank you for pointing these out. We will revise them in our camera-ready paper.
> >
> > [1] Laurent Dinh et al. NICE: Non-linear independent components estimation.
> >
> > [2] Laurent Dinh et al. Density estimation using real NVP.
> >
> > [3] Diederik Kingma et al. Glow: Generative flow with invertible 1x1 convolutions.
> >
> > [4] Chidubem Arachie et al. A general framework for adversarial label learning.
> >
> > [5] Lucas Theis et al. A note on the evaluation of generative models.
> >
> > [6] Jonathan Ho et al. Flow++: Improving flow-based generative models with variational dequantization and architecture design.
> >
> > [7]  Xuelin Chen et al. Unpaired point cloud completion on real scans using adversarial training.
> >
> > [8] Rundi Wu et al. Multimodal shape completion via conditional generative adversarial networks.

---

> > > ### Comment · Reviewer_W3Vi · 2021-11-23
> > > **Response to practicality and other technical comments**
> > >
> > > 1. Re: Practicality 1.  I don't think it is a sufficient to argue that because a previous work made a similar assumption that this work did, that it means the assumption is practically justified.  Further, while the ALL paper does indeed does some experiments to show their approach performs with noisy estimates, which helps understanding the practical utility of their method, I don't see similar experiments here.
> > >
> > > 2. Re: Practicality 2. I'm not sure this response directly addressed my feedback, but that could be because my feedback was not clear.  Stated another way: Can you give me real-world examples, based on prior work, where humans are easily able to provide rules that segment populations into real-valued targets?  The example used in the paper seems (to me) better suited for classification (e.g. If lab test A is > x then they have a disease, otherwise they do not).  If instead they are asked to provide, say, certainty estimates or rates of disease to be used as regression estimates, it's been shown that humans find this difficult [1], which led some works ([2] for example) to explicitly avoid using them as regression labels.  Given this, I would like to see counter examples where it is shown that that humans can provide real-valued labels with sufficient accuracy to learn from in this way.
> > >
> > > 3. Re: Technical Questions 1. I am not sure I understand why this is.  The most naive way would be sample uniformly over y and then reject samples that violate a constraint.  In principle, this seems possible.  Why is it not?  This is particularly important for my decision.
> > >
> > > 4. Re: Technical Questions 2. Understood.
> > >
> > > 5. Re: Technical Questions 3. Just to make sure I understand the response.  The numbers you report are close to those reported in prior work for the same data sets when they perform hyperparameter tuning?  I feel it is important to my decision to be convinced the evaluation is indeed fair and reflects realistic learning settings.
> > >
> > > [1] O'Hagan, Anthony, et al. "Uncertain judgements: eliciting experts' probabilities." (2006).
> > > [2] Nguyen, Quang, Hamed Valizadegan, and Milos Hauskrecht. "Learning classification with auxiliary probabilistic information." 2011 IEEE 11th International Conference on Data Mining. IEEE, 2011.

---

> > > > ### Author Response · Authors · 2021-11-29
> > > > **Response to Reviewer W3Vi**
> > > >
> > > > Thank you for your insightful comments and your clarification.
> > > >
> > > > 1. About the Practicality 1. A better way of saying what we meant is that we agree with the justification provided in previous work. Human experts can make guesses about error rate ranges, and error rates can also be obtained through other methods. That said, practicality is subjective, so our opinions may not align. We agree that it will be good to show experiments with different estimates of the error rates. We have performed these experiments and will include them in the appendix. Using the noisy error rates, LLF outperforms the baselines. Thanks for the suggestion.
> > > >
> > > >
> > > >
> > > > 2. About the Practicality 2. While the book [1] argues the case that it is difficult for humans to easily extract expert knowledge about certain unknown quantities, we don’t need exact estimates for weakly supervised learning. We still disagree that it is impractical to expect humans to provide weak signals about continuous values. It seems clear that human experts have some knowledge about average quantities. It would be surprising for a medical doctor to not know the average blood pressure for healthy 20–30 year old adults.
> > > >
> > > >
> > > > 3. About using uniform sampling. Uniform sampling is theoretically correct, but is impractical due to the curse of dimensionality. For the weakly supervised classification, the sample space of $\mathbf{y}$ is very big. For example, suppose we have $100$ data points, our sample space will have $2^100$ different samples of $\mathbf{y}$ vector. Among these samples, only a few of them satisfy the constraints. Therefore, the rejection rate is very high. In training, we will need to sample thousands of times in order to find the correct $\mathbf{y}$ vector, which will largely impact the training speed. For the unpaired point cloud completion problem, we want to generate $2048$ 3D points for each partial point cloud, and the points’ coordinates are continuous, but the high dimensionality again causes high rejection rates.
> > > >
> > > > 4. About the experiments. For the experiments of weakly supervised classification, we do not tune hyper-parameters for any models, including LLF and other baselines such as ALL. The baseline methods we used have published code online. They also provide default parameter settings for their models. We run their models with their provided default settings of parameters. Specifically, since we use the same setting as ALL, we use the code provided by ALL to process datasets and generate weak signals.
> > > >
> > > >     We would also like to mention that for weakly supervised classification, this is a tradition that we don’t tune hyper-parameters on validation sets. Previous works e.g., [2,3,4,5], all conduct experiments in this way, and they also cannot guarantee their methods achieve their best performance. We use other researchers’ code and parameter settings, in order to guarantee that other baselines can perform as good as they can.
> > > >
> > > > [1] O'Hagan, Anthony, et al. Uncertain judgements: eliciting experts' probabilities
> > > >
> > > > [2] Chidubem Arachie et al. A general framework for adversarial label learning
> > > >
> > > > [3] Alessio Mazzetto et al. Semi-supervised aggregation of dependent weak supervision sources with performance guarantees.
> > > >
> > > > [4] Alessio Mazzetto et al. Adversarial Multi-Class Learning under Weak Supervision with Performance Guarantees.
> > > >
> > > > [5] Chidubem Arachie et al. Constrained labeling for weakly supervised

---

> > ### Comment · Reviewer_W3Vi · 2021-11-23
> > **Response to clarification questions**
> >
> > This largely addresses my concerns about clarification, as long as the clarifications made in the comment are included (even in a small way) into the paper.

---

> > > ### Author Response · Authors · 2021-11-29
> > > **Response to Reviewer W3Vi**
> > >
> > > Thank you for your comments! We will add them to our paper.

---

### Official Review · Reviewer_fLEx · 2021-11-02

**Correctness:** 2
**Technical Novelty And Significance:** 3
**Empirical Novelty And Significance:** 2
**Recommendation:** 5
**Confidence:** 4

**Main Review:**

Section 3 gives a theoretical underpinning to the procedure, claiming that learning LLF is analogous to dequantization, i.e., learning LLF optimizes the likelihood of dequantized true labels. While I can see the similarity in Theorem 1, I fail to see how this motivates the use of LLF as an appropriate method for learning the true label's generating process, which ultimately should be our target in weakly supervised learning. Dequantization is typically considered to simplify the fitting of discrete observations, whereas in this paper we do not have any observations from the true generating process.

Section 4 gives instantiations of LLF to three different scenarios: weakly supervised classification, weakly supervised regression, and unpaired point cloud completion. The formulation for classification is from previous work (Arachie & Huang, 2021), with LLF as the solution instead of min-max optimization (Equation 8). For regression, the paper assumes the existence of M rule-based weak signals (Equation 9), which again enter the optimization problem as constraints (Equation 10). Lastly, the formulation for unpaired point cloud completion is presented, which is set-up differently from the classification and regression cases. It is more similar to a VAE-GAN formulation (Larsen et al., 2016) where the encoder is the conditional normalizing flow. Alternating steps are required during training as per traditional GAN.

### Pros

- Creating a unified framework to solve both (inaccurate) weakly supervised classification and regression is an important research question as collecting data sampled from the true data generating process can be an expensive process and while a lot of research has been done on solving weakly supervised classification, the regression case is under-studied. Having one paradigm can reduce the entry barrier for users who have to solve both problems.

- Formulating weak supervision, where there is no access to true labels, as an inverse flow which sidesteps the need to estimate the labels prior to training the model is an interesting formulation. This work makes use of the same weak supervision setting as ALL (Arachie & Huang, 2021), but the training is simpler as it does not require min-max optimization.

### Cons

- In unpaired point cloud completion, the complete clouds, which are true observations from the true data generating process, are considered as weak supervision, and the partial clouds are the covariates that we seek to map to being complete. In typical weak supervision scenarios, weak signals are generated in a paired manner to their covariates. Even though I can see how Equation 12 can be cast as Equation 6, I fail to see how unpaired point cloud completion is a weakly supervised learning problem. I was wondering if we can reformulate this problem such that the labels that are being used are truly weak, i.e., not coming from the true conditional p(y|x)? Moreover, just like the typical weakly supervised learning set-up, we should not tune on true validation sets.

- The paper states "However, these methods ignore this uncertainty between x and y", and "LLF uses a conditional flow to define the conditional distribution p(y|x), so that can model the uncertainty between input x and all possible y". However, the experiments only measure test set accuracies and not goodness-of-uncertainty metrics like NLL or calibration. If we want to make this claim, can we show that LLF performs good uncertainty quantification?
We are comparing approaches that only use weak signals against LLF (and ALL) which require the error rate bounds of the weak signals, an extra piece of information. In this paper, it is not clear to me how the error rate bounds are computed — in order to keep things fair, error rate bounds should be determined solely based on the weak signals. Is it possible to clarify this?

- Inference requires sampling and summarizing which can be slow and limits the use of LLF as a final predictor. On the other hand, two-stage approaches like Data Programming (DP) uses the label model to output probabilistic labels, which is then used to train a downstream neural network. As it is right now, if we compare their inference speeds, DP seems to be better. Can we also use LLF in a two-stage manner like in DP? How does it perform then?

### Additional Points for Discussion:
- Can you handle the case where you know of the dependence/correlational structure between the weak signals?

### Additional Experiments:
See Cons


**Summary Of The Paper:**

This article proposes **label learning flows (LLF)**, a general framework for weakly supervised learning tasks. The modelling framework is primarily driven by 1) conditional normalizing flows (Trippe & Turner, 2018), which is used as a flexible conditional likelihood model, with an affine coupling flow layer, and 2) constrained optimization formulation of maximum likelihood, where weak signals/error rate bounds enter as constraints (Arachie & Huang, 2021). The main practical formulation uses Lagrange multiplier to define an unconstrained stochastic loss function that requires Monte-Carlo samples during training (Equation 6). Ultimately, they obtain a trained inverse flow, which conditional on a covariate, can produce prediction samples.

**Summary Of The Review:**

Conceptually, constrained flow formulation of LLF is interesting and novel. However, its numerical results are not too convincing and there are some questions related to the practicality of this method.  I also do not think that unpaired point cloud completion should be an example of weakly supervised learning.  As it stands right now, I do not think that this paper meets the bar of "powerful and effective tool for weakly supervised learning problems". The paper can be much improved (especially for weak supervision practitioners audience) by addressing the questions under Cons and additional points above.

---

> ### Author Response · Authors · 2021-11-14
> **Response to Reviewer fLEx**
>
> Thank you for your insightful comments!
>
> 1. About the point cloud completion problem. Based on [1], incomplete supervision, inexact supervision, and inaccurate supervision are all weakly supervised. The unpaired point cloud completion problem lies in the category of inexact supervision problems. That is, we only have coarse labels, i.e., the referred set of complete point clouds, but don’t have exact labels, i.e., each partial point cloud does not have its corresponding complete point cloud. We agree that unpaired point cloud completion is not a typical weakly supervised learning problem, but we argue that it can be categorized as a weakly supervised learning problem with inexact supervision.
>
>     Our problem formulation follows exactly previous works [2,3], which allows us to use their methods as baselines, so that the comparisons are more fair. We agree that there may be other ways to define weak supervision for this problem. However, it may be difficult to develop other baselines for this reformulate problem.
>
>     In this problem, the datasets contain official validation sets, and the baseline methods [2,3] also tune model parameters on validation sets. For fair comparison, we also use the validation sets. Also, the validation sets don’t have paired partial and complete point clouds, and we only tune the parameters using TMD, MMD, and UHD as metrics.
>
> 2. About measuring the uncertainty. We agree that our experiments don’t verify that the uncertainty modeling of the probabilistic flow is the reason for better performance. One main reason is other baselines are deterministic models, so comparing their likelihoods is a little unfair. Besides, modeling uncertainty with a probabilistic model is our intuition for how we designed this framework. We actually did not claim that modeling uncertainty is the main reason for improving model performance. We will clarify this statement in our camera-ready paper.
>
> 3. About the experiment setting of weakly supervised classification. Due to the space limit, we put the details of experiment setting in appendix C.2. In a nutshell, our experiments of weakly supervised classification use the same methods as [4,5] to generate weak signals. For tabular datasets, we generate the weak signals and error bounds by training weak classifiers on simulation sets. We will clarify it in the main paper.
>
> 4. About the inference. LLF uses a normalizing flow to generate labels, so the inference process of LLF is very fast. The two-stage method uses a neural network to make final predictions for new data points. The prediction time for a neural network (if the network is reasonably big) should be similar to the sampling time of LLF. Besides, we can also use LLF to generate labels for training down-stream neural networks. However, since LLF itself uses a neural network to make predictions, training another neural network is a little redundant.
>
> 5. Additional points. The conditional flow $p(\textbf{y}|\textbf{x})$ itself does not affect the dependencies among weak signals. To model dependencies among weak signals, we would need to define new constraints specifically that consider the relationship between the error rates across known dependent signals. For now, other constraint-based weak supervision methods also do not exploit this information, and it would be an interesting direction for future work.
>
> [1] Zhi-hua Zhou. A brief introduction to weakly supervised learning.
>
> [2]  Xuelin Chen et al. Unpaired point cloud completion on real scans using adversarial training.
>
> [3] Rundi Wu et al. Multimodal shape completion via conditional generative adversarial networks.
>
> [4] Chidubem Arachie et al. A general framework for adversarial label learning
>
> [5] Chidubem Arachie et al. Constrained labeling for weakly supervised learning.

---

> > ### Comment · Reviewer_fLEx · 2021-11-19
> > **Response to authors (1/2)**
> >
> > > The unpaired point cloud completion problem lies in the category of inexact supervision problems. That is, we only have coarse labels, i.e., the referred set of complete point clouds, but don’t have exact labels, i.e., each partial point cloud does not have its corresponding complete point cloud.
> >
> > Thanks for the paper reference! I was actually using this paper to land on my initial point.  I had attempted to fit the point cloud completion into one of the three weak supervision definitions based on that paper, but I did not manage to, since all the definitions seem to require that all the (weak) labels are explicitly associated to some input X.
> > In inexact supervision, we still want the coarse-grained label to be *associated* with an input X.  In that paper’s example, object (in image) categorisation where the label (category) is available for the whole image is mentioned as an example — here the coarse-grained image-level label is associated to its respective object/image.  A complete point cloud does not seem to be associated to any input partial point cloud.
> >
> > > We actually did not claim that modeling uncertainty is the main reason for improving model performance. We will clarify this statement in our camera-ready paper.
> >
> > Noted, thank you.
> >
> > > About the inference. LLF uses a normalizing flow to generate labels, so the inference process of LLF is very fast. The two-stage method uses a neural network to make final predictions for new data points. The prediction time for a neural network (if the network is reasonably big) should be similar to the sampling time of LLF. Besides, we can also use LLF to generate labels for training down-stream neural networks. However, since LLF itself uses a neural network to make predictions, training another neural network is a little redundant.
> >
> > Thanks for your response, I just wanted to clarify my initial point — I wanted to compare the inference speed of a trained neural network (the final product of Data Programming)  and that of evaluating multiple inverse flows (sampling) and averaging (LLF).  Principally, it seems like the former (pointwise) is much faster than the latter (Monte-carlo like), and this seems to be strengthened by what you mentioned “LLF itself uses a neural network to make predictions”.
> >
> > I believe that you can treat LLF as a “label model” (in DP parlance) and generate probabilistic weighted examples (from your sampling) which are used to train a downstream NN, in order to match the computational complexity of a two-stage Data Programming.  But in that case, I think we ought to evaluate the downstream NN’s predictive performance and not the MC samples from the inverse flow.

---

> > > ### Author Response · Authors · 2021-11-23
> > > **Response to Reviewer fLEx**
> > >
> > > Thank you for your clarification and your insightful comments!
> > >
> > > 1. About the unpaired point cloud completion. Thank you for your clarification. We can also interpret the problem in this way. In Equation 12, one constraint is from the discriminator of GAN, i.e., $D(\mathbf{g}_{\mathbf{x}_p}(\mathbf{z}))$. Note that this discriminator is trained on the set of referred complete point clouds and contains the structure and geometric information of complete point clouds. When we train the conditional flow, we fix this discriminator. The generated complete point cloud $\mathbf{y} = \mathbf{g}( \mathbf{z} )$ will be input to this discriminator and output a score within $[0,1]$ to represent how realistic the generated point cloud is. This score is explicitly related to the input partial point cloud $\mathbf{x}_p$, and it restricts the $\mathbf{y}$ to have similar structure and shape as the real complete point clouds. Therefore, this score output from the discriminator can be roughly seen as structure information of each input partial point cloud, which is an instance-wise weak supervision.
> > >
> > >    More details about LLF for partial point completion can be seen in appendix C.1.3. We will clarify this problem in our camera-ready paper.
> > >
> > >
> > > 2.  About the inference. Thank you for your clarification. We agree that LLF can be used as a “label model” for a two-stage method. In this situation, we will need to compare the performance of downstream neural networks.
> > >
> > >      In terms of the inference time of LLF, we use $10$ samples to estimate labels in our experiments, and these $10$ samples can be generated simultaneously, so the inference of LLF is also very fast, i.e., with a high-performance GPU, generating $10$ samples together is as fast as generating $1$ sample, due to the GPU’s ability of parallel computing. The inference time of LLF and that of a supervised neural network largely depend on the network sizes, GPU speed, and programming method, so it is hard to make a direct comparison.
> > >
> > > 3. About the error bounds. We split a dataset to training set, simulation set and test set. We assume that simulation sets have true labels. We train weak classifiers on simulation sets and use them to generate weak signals. We also use the true labels in simulation sets to compute error bounds. This simulation method is the same as that used in ALL[4]. In their original paper, they call the simulation sets “validation sets”. However, we feel that the term “simulation set” should be more appropriate, since these datasets are used for simulating the process of weak signal generation rather than validating trained models. In our experiments, LLF and ALL use the same error bounds. Other baselines do not require error bounds. All methods use the same weak signals, so our experiments are fair. We will clarify it in our camera-ready paper.
> > >
> > > 4. About Theorem 1. You are right that Theorem 1 is not related to the motivation of using normalizing flows. Theorem 1 only gives an intuition of the similarities between the training of LLF and the dequantization technique. We include this theorem in order to help readers to better understand the training of LLF, since it is different from the training of traditional flows.
> > >
> > >    One main motivation of using normalizing flows to represent the conditional distribution $p(\mathbf{y}|\mathbf{x})$ is that in training, we can generate the labels $\mathbf{y}$ and compute the objective function $\log p(\mathbf{y}|\mathbf{x})$ together, so training LLF is as easy as training other neural networks, i.e., it only requires using back-propagation to compute gradients. If we use other methods, e.g., logistic regression, to represent $p(\mathbf{y}|\mathbf{x})$, the training will become an EM-like algorithm, i.e., we first estimate $\mathbf{y}$ by optimizing the constraints $\mathbf{C}$, and then learn the parameters of $p(\mathbf{y}|\mathbf{x})$ using the estimated $\mathbf{y}$. ALL[4] uses this EM-like algorithm to train their models. We will clarify our motivations in our camera-ready paper.

---

> > > > ### Comment · Reviewer_fLEx · 2021-11-24
> > > > **Second response to authors**
> > > >
> > > > > and output a score within [0,1] to represent how realistic the generated point cloud is. This score is explicitly related to the input partial point cloud xp, and it restricts the y to have similar structure and shape as the real complete point clouds
> > > > > Therefore, this score output from the discriminator can be roughly seen as structure information of each input partial point cloud, which is an instance-wise weak supervision.
> > > >
> > > > Thank you for this elaboration. It seems to me that the discriminator is trained using true generating process data (complete point clouds).  In that case, I'm not sure where the weak supervision is.  Another way to check this elaboration is to figure out what kind of weak supervision (out of the three types) this is the discriminator output for x_p, it does not seem to be one of the three categories.
> > > >
> > > > The casting argument you presented also seems to imply that GAN models where the discriminator is trained using some reference dataset can be generally cast as a weak supervision problem.  I think this general implication deserves more discussion and elaboration.
> > > >
> > > > > We agree that LLF can be used as a “label model” for a two-stage method. In this situation, we will need to compare the performance of downstream neural networks.
> > > > > In terms of the inference time of LLF, we use 10 samples to estimate labels in our experiments, and these 10 samples can be generated simultaneously, so the inference of LLF is also very fast, i.e., with a high-performance GPU, generating samples together is as fast as generating sample, due to the GPU’s ability of parallel computing.
> > > >
> > > > While I am sure that given enough GPU resources one can always equalize the wall-clock time of NN inferences+averaging (LLF) and NN inference (distilled downstream NN, typical of two-stage models), the former still requires more compute resources.   When compute resources are scarce and there is a demanding SLA in inference, it may be difficult to use the former, and two-stage-style downstream neural networks must be used.
> > > >
> > > > > We also use the true labels in simulation sets to compute error bounds. This simulation method is the same as that used in ALL[4].
> > > >
> > > > Does this mean that the true labels are used as an extra component (via error bounds, and thus, the constraints) for LLF and ALL (aside from the weak signal generation)? If that's the case, in Table 1, it seems fitting that they are better than GE and AVG, and in some cases (Fashion MNIST DvK/Statlog Satellite), the performances are almost equal or equal to supervised learning's result.
> > > >
> > > > Similarly, if the true labels are used as an extra component for LLF in Table 2, it seems fitting that it's performing the best.  While this helps illustrate the idea, care must be taken to make sure that the usage of true labels across the different methods are equivalent (as much as possible. For example, if the DP is compared against it, the labelling functions in the DP should be of the same level of correctness/misspecification as the weak signals and bounds used by LLF.  It would be great to include these baseline details, at least in the appendix.
> > > >
> > > > Moreover, in reality we do not have access to true labels, and therefore we need to compute the error bounds using other means -- would be great if this can be clarified in the paper for practitioners who are looking to use LLF.
> > > >
> > > > > Theorem 1 only gives an intuition of the similarities between the training of LLF and the dequantization technique. We include this theorem in order to help readers to better understand the training of LLF
> > > >
> > > > Noted, thanks for the clarification and for the main motivation elaboration.

---

> > > > > ### Author Response · Authors · 2021-11-29
> > > > > **Response to Reviewer fLEx**
> > > > >
> > > > > Thank you for your insightful comments.
> > > > >
> > > > > 1. About the unpaired point cloud completion. Based on the definition in [1]:  ''Inexact supervision concerns the situation in which some supervision information is given, but not as exact as desired.''
> > > > >
> > > > >     We therefore categorize unpaired point cloud compilation as an inexact supervision problem. This is because in this problem, we want to generate complete point clouds $\mathbf{y}$, but we don’t have paired data points in our datasets. We use the GAN discriminator to restrict $\mathbf{y}$, therefore, the score output from the discriminator is a kind of inexact supervision.
> > > > > Ground truth labels can also be used as weak supervision for training, if they are not directly related to the $\mathbf{y}$ we want to predict. For example, in weakly supervised object localization[2], we use image level labels as weak supervision, and the task is to predict pixel-wise labels. In relation to our work, we will not always have the complete point cloud for certain examples as part of the reference set. Hence why we classify it as inexact supervision.
> > > > >
> > > > >
> > > > > 2. We agree that when we only have limited computing resources, the two-stage method will be more efficient. We will add this discussion to our updated paper.
> > > > >
> > > > > 3. In our experiments, LLF and ALL use the same error bounds, since they actually use the same constraints. We agree with your argument that the error bounds contain some information of true labels, so it is a possible reason why these methods outperform other baselines. We will add discussions about the error bounds and the details of how to train other baselines to our updated paper.
> > > > >
> > > > >     We have new experiments using noisy estimates of the error rates, and LLF outperforms DP in the results. This is consistent with the experiments in ALL where using incorrect estimates of the error rates only slightly impacts the results. We will add these new experiments in the appendix and provide more clarification about our experiments set up.
> > > > >
> > > > >
> > > > > [1] Zhi-hua Zhou. A brief introduction to weakly supervised learning.
> > > > >
> > > > > [2] Junsuk Choe. Evaluating Weakly Supervised Object Localization Methods Right.

---

> > ### Comment · Reviewer_fLEx · 2021-11-19
> > **Response to authors (2/2)**
> >
> > > In a nutshell, our experiments of weakly supervised classification use the same methods as [4,5] to generate weak signals. For tabular datasets, we generate the weak signals and error bounds by training weak classifiers on simulation sets. We will clarify it in the main paper
> >
> > Thank you for referring me to [4,5].  I have checked them, and I am still wondering about the error bounds — please correct me if I’m wrong, but in these past works, the error bounds are generated either using the true clean data (and the test results used more as an oracle) or guessed arbitrarily.
> > In this paper, you’ve written “and estimate error bounds and thresholds on the simulation set.” — how exactly are the error bounds estimated from training weak classifiers on the simulation set and what’s a simulation set? I might have missed it, but I could not find “simulation set” in [4,5].  Are there labels in the simulation set and are they clean labels from the true data generating process or are they already made into weak labels before you use them for estimation?   In order for comparisons against other methods to be fair, I believe that the error bounds *must not be* estimated from labels sampled from the true data generating process.
> > I think it is important to include details for this set-up even though there are similar set-ups in [4,5], since LLF's methodology is very different from from the past ones.
> >
> > > To model dependencies among weak signals, we would need to define new constraints specifically that consider the relationship between the error rates across known dependent signals. For now, other constraint-based weak supervision methods also do not exploit this information, and it would be an interesting direction for future work.
> >
> > Thank you for clarifying.
> >
> > There was another comment I made that I would like a clarification on but it was not under cons. I apologise for not being clear about this.  My comment:
> > > Section 3 gives a theoretical underpinning to the procedure, claiming that learning LLF is analogous to dequantization, i.e., learning LLF optimizes the likelihood of dequantized true labels. While I can see the similarity in Theorem 1, I fail to see how this motivates the use of LLF as an appropriate method for *learning the true label’s generating process, which ultimately should be our target in weakly supervised learning*. Dequantization is typically considered to simplify the fitting of discrete observations, whereas in this paper we do not have any observations from the true generating process.

---

### Official Review · Reviewer_ii99 · 2021-11-03

**Correctness:** 3
**Technical Novelty And Significance:** 3
**Empirical Novelty And Significance:** 3
**Recommendation:** 6
**Confidence:** 3

**Main Review:**

Strength:

1. This paper presents a new method based on normalising flow for weakly supervised learning problems. It seems to be novel in this specific setting.

2. The paper has a comprehensive set of experiments in a variety of applications.

3. Overall, the paper is well written and easy to follow.

Weakness:

1. It is a bit unclear why normalising flow is used. Specifically, in the weakly supervised learning setting of the paper, $y$ is unobserved but we know a set of constraints of $y$, i.e., $C$. It is unclear why directly parameterising $\log{p(y|x)}$ and optimising with $C$ is impossible and why normalising flow makes it possible. I feel that the context around Eq (4) needs more explanation.

2. It seems that there are not many existing methods that exactly follow the setting of the paper. Therefore, it's better to have a more comprehensive comparison with the latest related methods, such as PGMV and its variants in [1] and AMCL and its variants in [2].

[1] Mazzetto, Alessio, et al. "Semi-supervised aggregation of dependent weak supervision sources with performance guarantees." International Conference on Artificial Intelligence and Statistics. PMLR, 2021.

[2] Mazzetto, Alessio, et al. "Adversarial Multi-Class Learning under Weak Supervision with Performance Guarantees." International Conference on Machine Learning. PMLR, 2021.

**Summary Of The Paper:**

This paper proposes a new method for weakly supervised learning where the true label of a data sample is unknown but there are known constraints on the label. The proposed method is based on conditional normalising flows, where the inverse flow is used to model the conditional label distribution and optimised with the label constraints. The proposed method is used in three applications including classification, regression, and point cloud completion. The proposed method is shown to have better performance in these applications in the comparison with several baseline methods.



**Summary Of The Review:**

The idea of the paper is interesting and seems to be effective. But more explanations and comparisons are needed.

---

> ### Author Response · Authors · 2021-11-14
> **Response to Reviewer ii99**
>
> Thank you for your insightful comments!
>
> 1. About the motivation of using normalizing flows. The main problem in training is we want to optimize the conditional likelihood $\log p(\textbf{y}|\textbf{x})$. However, the $\textbf{y}$ is unobserved. We make use of the advantage of normalizing flow and develop a training method, i.e., Equation 4 and Equation 5, which trains the flow inversely, and avoids the need of observed $\textbf{y}$. Therefore, in training, we generate $\textbf{y}$ and optimize the objective function simultaneously. We may use other methods, e.g., logistic regression, to define $p(\textbf{y}|\textbf{x})$. However, in this way, we will need to first estimate $\textbf{y}$ by optimizing the constraints, and then optimize $\log p(\textbf{y}|\textbf{x})$ with this estimated $\textbf{y}$, resulting in an EM like algorithm, e.g., ALL[1]. We will clarify our motivation in our camera ready paper.
>
> 2. About the experiment setting. You are correct that we can make additional comparisons with more papers in this line of work. However, these papers [2,3] are very new and we were wrapping up our experiments at the time they were published. We did include discussion about them in the related work section of our paper. It is also worth mentioning that these papers are variants of the ALL method and we compare against CLL [4] which is also a new variant of ALL.
>
> [1] Chidubem Arachie et al. A general framework for adversarial label learning
>
> [2] Alessio Mazzetto et al. Semi-supervised aggregation of dependent weak supervision sources with performance guarantees.
>
> [3] Alessio Mazzetto et al. Adversarial Multi-Class Learning under Weak Supervision with Performance Guarantees.
>
> [4] Chidubem Arachie et al. Constrained labeling for weakly supervised learning.

---

### Official Review · Reviewer_mY6j · 2021-11-03

**Correctness:** 4
**Technical Novelty And Significance:** 3
**Empirical Novelty And Significance:** 3
**Recommendation:** 8
**Confidence:** 4

**Main Review:**

Overall, I like the paper and vote to accept. The conversion of the traditional use of normalizing flows is clever and appears to be effective. I really appreciate the variety of the experiments, e.g., a classification problem, a regression problem, and a generative problem. That the generative problem is additionally performed within the constraints of point clouds/sets is very impressive and really helps to illustrate the versatility of the idea.

That said, I had to read section 4 several times before I understood how the constraints were constructed for a particular problem and am not sure I could construct them on a different problem set. The paper would be considerably more accessible with a simple toy problem with figures/cartoons. This could also be used to help better motivate "weak signals."

I'm also curious why the paper (exclusively?) utilizes affine coupling layers to construct the normalizing flow when more advanced methods exist (e.g., spline coupling).
The use of a GAN as part of the constraints alongside the flow-model appears to be a bit of a contradiction but I think helps to illustrate the variability that is allowed within the framework.

**Summary Of The Paper:**

The paper proposes to utilize conditional normalizing flows to exploit weak supervisory signals by converting the conventional likelihood maximization problem into a constrained optimization problem. The paper develops three (very) different constrained problems and demonstrates the method on each.

**Summary Of The Review:**

The paper proposes a clever twist on conditional normalizing flows and shows the versatility of the idea on several different weakly supervised problems. An illustration would help to make the paper more accessible which would increase it's impact.

---

> ### Author Response · Authors · 2021-11-14
> **Response to Reviewer mY6j**
>
> Thank you for your positive comments!
>
> 1. About the toy examples. Your suggestions are very helpful! We will add some diagrams to illustrate weak signals and our LLF in our camera-ready paper.
>
> 2. About constructing constraints. Developing constraints with weak signals for labels is problem-specific. It largely depends on the weak signals and the problem itself. We show two common forms of constraints: error constraints and average value constraints, but there are likely many other possible ways to encode the weak supervision in different settings.
>
> 3. About the affine coupling layers. Since our variables are vectors, we don’t need to use invertible linear layers, e.g., 1x1 convolution [1], which are developed basically for high-dimensional tensors such as images. In terms of coupling layers, we choose conditional affine coupling layers because they have been widely used in conditional flows, e.g., [2,3,4]. As you suggest, more advanced coupling layers may be possible by designing conditional versions, and we can incorporate this into our framework in future work.
>
>
>
> [1] Diederik Kingma et al. Glow: Generative flow with invertible 1x1 convolutions.
>
> [2] Roman Klokov et al. Discrete point flow networks for efficient point cloud generation.
>
> [3] You Lu et al. Structured output learning with conditional generative flows.
>
> [4] Albert Pumarola et al. C-flow: Conditional generative flow models for images and 3d point clouds.

---

### Decision · Program_Chairs · 2022-01-20

**Decision:**

Reject

**Comment:**

The paper proposes a new approach for weakly supervised learning, based on conditional normalizing flows. Reviewers generally found the paper to have an interesting, novel proposal with empirical promise. However, some concerns were raised: to name a few,

(1) _Clarity._ Several reviewers found portions of the technical content hard to follow, e.g., the description of constraints in Sec 4.

(2) _Scalability compared to data programming._ One reviewer was unsure of how the present approach compares in terms of inference time and/or accuracy to a two-stage data programming approach.

(3) _Infeasibility of sampling from Equation 2._ One reviewer suggested the paper discuss and compare to a simpler baseline, which is to perform rejection sampling from the constraint set.

(4) _Suitability of point cloud problem._ One reviewer was unsure of whether the point cloud problem, considered as an experimental setting in this paper, is reflective of weakly supervised learning.

(5) _Practicality of knowing weak labeler error rates._ The paper assumes knowledge of the weak labeler error rates in constructing constraints. Some reviewers raised concerns on the practical viability of this assumption.

For point (2), the relevant reviewer was not convinced following the discussion. The suggestion is to treat LLF as a label model, which serves as input to a non-MC predictor. The question then is what the predictive performance of this combined approach looks like, as opposed to the LLF's themselves.

For point (3), the response clarified that the number of constraints might make rejection sampling infeasible. This appears to be true, but it is suggested that the paper at a minimum discuss this, and ideally also clarify claims about the general-purpose need for the proposed approach (since in some cases one might be able to do rejection sampling).

For point (4), the discussion was somewhat inconclusive. It is suggested that the authors explicitly discuss some of the points brought up in the response.

For point (5), while the assumption not wholly uncommon in the literature, it would be better for the authors to perform some sensitivity analysis against misspecification of the error rates.

Overall, the paper has some interesting ideas that are well worth exploring. The present execution appears to have some scope for improvement, with the reviews providing a range of suggestions of areas of the paper that could be made clearer or strengthened. The paper would be best served by incorporating these comments and undergoing a fresh review.